# In Vivo Analysis of Optic Fissure Fusion in Zebrafish: Pioneer Cells, Basal Lamina, Hyaloid Vessels, and How Fissure Fusion is Affected by BMP

**DOI:** 10.3390/ijms21082760

**Published:** 2020-04-16

**Authors:** Priska Eckert, Max D. Knickmeyer, Stephan Heermann

**Affiliations:** 1Department of Molecular Embryology, Institute of Anatomy and Cell Biology, Faculty of Medicine, University Freiburg, 79104 Freiburg, Germany; Priska.Eckert@anat.uni-freiburg.de (P.E.); Max.Knickmeyer@anat.uni-freiburg.de (M.D.K.); 2Faculty of Biology, University of Freiburg, Schaenzlestrasse 1, D-79104 Freiburg, Germany

**Keywords:** BMP, optic fissure, basal lamina, POM, pioneer cells, hyaloid vessel

## Abstract

Colobomata, persistent optic fissures, frequently cause congenital blindness. Here, we focused on optic fissure fusion using in vivo time-lapse imaging in zebrafish. We identified the fusion initiating cells, which we termed “pioneer cells.” Based on morphology, localization, and downregulation of the neuroretinal (NR) precursor marker *rx2*, these cells could be considered as retinal pigment epithelial (RPE) progenitors. Notably, pioneer cells regain *rx2* expression and integrate into the NR after fusion, indicating that they do not belong to the pool of RPE progenitors, supported by the lack of RPE marker expression in pioneer cells. They establish the first cellular contact between the margins in the proximal fissure region and separate the hyaloid artery and vein. After initiation, the fusion site is progressing distally, increasing the distance between the hyaloid artery and vein. A timed BMP (Bone Morphogenetic Protein) induction, resulting in coloboma, did not alter the morphology of the fissure margins, but it did affect the expression of NR and RPE markers within the margins. In addition, it resulted in a persisting basal lamina and persisting remnants of periocular mesenchyme and hyaloid vasculature within the fissure, supporting the necessity of BMP antagonism within the fissure margins. The hampered fissure fusion had severe effects on the vasculature of the eye.

## 1. Introduction

During eye development in vertebrates, specifically during optic cup morphogenesis, a fissure forms in the ventral part of the optic cup, termed optic fissure or choroid fissure [1]. This fissure is a physiological (but transient) gap [2,3] and is important during a specific period of development for the entry of periocular mesenchymal cells (POMs) and POM-derived embryonic vasculature (hyaloid vessels) [4]. It is essential that the optic fissure is closed as development proceeds. A persisting optic fissure is termed coloboma and is a major cause for blindness in children [5,6]. Next to some environmental and dietary factors, many genes have been linked to coloboma formation [7], resulting in a coloboma gene network [8,9] which recently was updated [6]. It consists of many signaling pathway components, e.g., of Wnt [10,11], fibroblast growth factor (FGF) [12,13], retinoic acid (RA) [14,15], Hippo [16], sonic hedgehog (Shh) [17], transforming growth factor β (TGFβ), and bone morphogenetic protein (BMP) [18,19].

Importantly, the morphology of coloboma phenotypes is highly variable, arguing at least for two main distinct coloboma classes [18,20]: morphogenetic coloboma, showing a vast cleft [1,11,16,21], and fusion defective coloboma, in which the margins were aligned but remained unfused [12,13,18].

The focus of our current analysis was the process of optic fissure fusion. A prerequisite for the two fissure margins to fuse is a regular architecture of the respective margins, meaning that the morphogenesis of the optic fissure must have occurred in an orderly manner [1]. Subsequently, when the fissure margins are aligned and in touch, the basal lamina has to be degraded [22,23]. A persistent basal lamina and thus coloboma could be found in Vitamin A–deficient rat embryos [24]. This effect was likely involving POM, supported by data showing that blocking retinoic acid (RA) signaling in neural crest derivatives resulted in coloboma [14]. However, it was also proposed that RA signaling is important within the POM and the optic cup independently to support optic fissure closure [15]. The importance of the POM for optic fissure fusion was further strengthened by transplantation experiments indicating that without POM, fissure fusion is not occurring [22]. The hyaloid vasculature, or potentially their precursors derived from POM, was found to be important for basal lamina degradation [23]. Interestingly, the lack of the early hyaloid vessels addressed in cloche mutants in which the vasculogenesis is impaired due to failure in endothelial specification resulting from a mutation of the master regulator *npas4l* [25,26,27,28] did delay, but not prevent, basal lamina dissolution [23]. The lack of the hyaloid vasculature had severe other effects, e.g., mis-patterning and microphthalmia; a coloboma, however, was not reported [29] (and own observations). On the other hand, it was described that an enlarged hyaloid vessel found in *lmo2* zebrafish mutants hampered fissure fusion [30], *lmo2* being an endothelial factor needed for angiogenesis rather than vasculogenesis as *npas4l*. This is interesting since *lmo2* was found to be regulated by *npas4l* [28,31]. Another recent study identified the expression of Netrin, a factor localized to the basal lamina, coinciding with its degradation [32]. Also Pax2 seemed to be involved in basal lamina degradation [33]. Notably, a loss, but also a gain of Pax2 results in coloboma, with a persistent basal lamina [34]. Furthermore, TGFβ was found important for basal lamina degradation [18]. Even though many factors were linked to the degradation of the basal lamina and recent transcriptional analyses include potential effector molecules [18,32,35], such factors need further analysis and validation. Besides, it is also not clear whether the POM and the NR/RPE precursors cooperate during basal lamina degradation.

Notably, BMP antagonists were found to be expressed in the optic fissure margins partially under the control of TGFβ, and timed overexpression of BMP4 was hampering optic fissure fusion [18], although the mechanism is still elusive. BMP signaling was furthermore found to play an important role during other steps of eye development. BMP is important for axis specification [36,37,38], the development of the optic fissure [1,39] and also for retinal pigment epithelial (RPE) development [40,41]. Notably, recent findings during optic cup morphogenesis [21] indicate a dynamic movement of the boundary between the future neuroretinal (NR) and RPE. This was also found to depend on BMP antagonists. These data underline the role BMP plays for RPE development but also suggest that tissue rearrangements during morphogenesis need to be considered. Moreover, defective BMP signaling was recently identified to be the cause of another morphogenetic defect of the eye—superior ocular coloboma [42]. This malformation affects the superior ocular sulcus, which can be described as a smaller dorsal optic fissure. A blood vessel growing through this sulcus was also found to develop aberrantly as a result of the morphogenetic defect.

During optic fissure fusion, next to the dissolution of the basal lamina, the morphology of the fissure margins has to change. The latter involves changes in cell polarity [22] and adjustment of cell–cell contacts. Among potentially many factors, FAT1, an atypical protocadherin, was found to be important for fusion [43]. In addition to these structural changes, a tight control of cell proliferation and cell death seems to be important to prevent coloboma [44,45,46]. The latter, however, might be linked to both morphogenetic coloboma and fusion coloboma.

Even though many coloboma-related genes are known, and scientific progress in the field has shed more light on the process of optic fissure fusion in recent years, some of its aspects still remain elusive. In the margins of the optic fissure, the precursors of NR and RPE are positioned next to each other, sharing a common basal lamina. After fusion, these two epithelia are separated. Notably, which of these cell types initiates the fusion has long remained anonymous. In an ultrastructural analysis of fissure fusion in mice [47], it was found that the cellular contact in between the margins is established between flattened cells. These cells, based on morphology, were called RPE precursors. This was an important finding; however, due to the focus on ultrastructure of the process in mice, the study could not provide information of the dynamics of the fusion process.

Here, we used time-lapse in vivo microscopy and zebrafish (*Danio rerio*) embryos to address how and where the cellular contact in between the margins is established. We focused on the putative cell type initiating the cellular contact, the dynamics of the process, and then consecutively followed up on the fusion domain. Furthermore, we investigated how induced BMP signaling is negatively affecting optic fissure fusion.

We identified the first cellular contact between the margins in the upper third of the optic fissure, the region in which we and others also found the onset of basal lamina dissolution. The cells, which mediated the contact between the margins, appeared to be RPE progenitors based on their localization, their morphology, and the downregulation of the neuroretinal precursor marker *rx2*—or rather, a *rx2* cis regulatory element [21]. These cells, which we termed pioneer cells, established the first cellular contact between proximal fissure margins and thus separated the hyaloid artery from the hyaloid vein. Unexpectedly, pioneer cells regained *rx2* expression and integrated into the NR after fusion, suggesting that they do not merely belong to the pool of retinal pigment cell progenitors. Further investigation of RPE and NR markers was supporting this finding. After initiation, close to the future optic nerve head, the fusion site was found to move distally. Thus, the distance between the hyaloid artery and vein was increasing. The fusion site itself was located in close proximity to the hyaloid vein. Moreover, we found that a timed BMP induction, resulting in colobomata [18], also resulted in a persisting basal lamina, suggesting that BMP antagonism within the fissure margins is important for basal lamina dissolution. BMP induction did not alter the morphology of the fissure margins but affected the expression of NR and RPE markers within the margins. The POM derived vasculature was found to be severely affected, likely as an indirect result of the failed fusion of the optic fissure.

Data concerning the first description of the pioneer cells, as a specific population of cells within the fissure margins, was pre-published [48]. This data was cited and discussed in recent literature [22,32].

## 2. Results and Discussion

### 2.1. Pioneer Cells Establish the Contact between the Optic Fissure Margins

We first addressed which cell population establishes the cellular contact between the optic fissure margins to initiate their fusion. To this end, we used in vivo time-lapse imaging of zebrafish embryos. It was shown previously that an *rx2* cis regulatory element of medaka (*Oryzias latipes*) drives expression initially in all retinal progenitor cells meaning progenitors for both NR and RPE and is subsequently restricted to progenitors of the NR [21]. We used the transgenic line *tg(Ola.rx2:eGFP-caax)* [21] and ubiquitous cell membrane labeling by mRNA injections of lyntdTomato into zygotes and recorded 3D confocal stacks over time from a lateral perspective (Figure 1A, oriented nasal to the left). The two-color labeling enabled us to discriminate the *rx2* negative population of prospective RPE progenitor cells from the *rx2* positive progenitor cells of the NR (Figure 1B,B’). We observed that the first cellular contact in between the fissure margins was established by cells with flattened morphology and a markedly reduced green fluorescence, driven by the aforementioned *rx2* cis regulatory element [21] (Figure 1B,B’,C,C’, Appendix A’,B,B’). Thus, these cells could be characterized as presumptive RPE precursors, in line with previous findings [47]. We named these cells “pioneer cells.” Unexpectedly, we observed that the *rx2* driven GFP expression (at that stage indicative for NR progenitor cells) was reactivated after contact formation, during subsequent stages of fissure fusion (Figure 1D,D’,E,E’; Appendix A). These findings indicate that the pioneer cells, even though they showed RPE progenitor features at the onset of contact formation, might not belong to the pool of RPE precursors but constitute a separate cell type that can gain NR progenitor features during fusion. After retinal differentiation, *rx2* is expressed in photoreceptors and Müller glia cells [49]. Notably, in the postembryonic ciliary marginal zone, *rx2* is indicative for retinal stem cells [49]. It is repressed in the adjacent transit amplifying zone, however, can be found re-expressed in photoreceptors and Müller glia cells [49]. Our data suggest an additional event during which *rx2* expression, followed here by using an *rx2* cis regulatory element driving GFP expression, is dynamic, including repression followed by a reactivation.

### 2.2. First Contact between Optic Fissure Margins Occurs Close to the Prospective Optic Nerve Head

Next, we addressed the exact location of the first cellular contact with respect to the total height and to the proximal-distal axis of the optic fissure. To this end, we determined the point of the cellular contact establishment using 3D stacks, recorded from a lateral perspective over time. Measurements were performed with Fiji [50]. The height of the optic fissure was estimated according to the scheme presented in Figure 2 (Figure 2A, arrow). The point of contact was then related to the height of the optic fissure (Figure 2A). Our data show that the pioneer cells establish the contact in the upper third of the fissure (Figure 2B,C). This matches previous observations of contact formation [47] and the site of basal lamina dissolution (Figure 2D–G’), also shown in previous studies [23]. This suggests that the boundary between the prospective NR and RPE, which is highly dynamic during morphogenesis [1,21], has to be brought into this position during optic fissure formation to enable the subsequent fusion of the margins. Therefore, it is plausible that failures of optic cup formation, resulting in an eversion of the boundary, result in coloboma [1]. It will be interesting to investigate in prospective analyses whether the pioneer cells are situated within the boundary before it arrives in the margins or are specified afterwards. Notably, the appearance of the site of fissure fusion is different in samples subjected to immunohistochemistry (Figure 2D–G’, Appendix A´´), with the fissure margins being in closer contact overall, likely due to a shrunken hyaloid vessel which increases the space between the margins in vivo (compare to Figure 1C,C’, asterisk, Appendix A, showing a comparison). Taking the information from our in vivo analyses and from our analyses on processed embryos into account, it seems likely that the pioneer cells are actively approaching each other with the basal lamina being still intact (Appendix A).

We next aimed to estimate the point of the first cellular contact between the fissure margins in relation to the proximal–distal axis. Measurements were performed using Fiji [50] and in vivo recordings of 3D stacks from a lateral perspective over time. We identified the point of the first contact formation (FC) and related this to the position of the future optic nerve head (ONH) and the proximal end of the forming lens (LP) (Figure 2G). In zebrafish, the space of the primary vitreous body in between the embryonic retina and the forming lens is considerably small. The median distance between the forming nerve head and the proximal end of the forming lens was 15 µm (Figure 2G, ONH-LP) at the time of contact formation. We next measured the distance between the site of first contact formation and either the prospective optic nerve head or the back of the lens. The median distances were 6 µm and 9 µm, respectively (Figure 3G, ONH-FC, FC- LP). Our data indicate that the fusion of the optic fissure is initiated close to the optic nerve head in zebrafish.

We further observed a progression of fissure fusion from proximal to distal regions, which in part had been previously observed [23,47,51]. Bernstein and colleagues described a similar progression of fissure fusion towards distal compartments in mice, chicks, and zebrafish but also described a fusion by intercalation in proximal regions of the chicken optic fissure [51]. Others reported the onset of fusion in between the proximal and distal ends with a progression of fusion toward both sides in mice, where an onset close to the future optic nerve head was also described [47]. In zebrafish, it was reported that the fissure fusion progresses toward both the proximal and distal directions [23,47], though this consideration potentially includes the optic stalk [22]. It is conceivable that the process described as fusion by intercalation is important for the definition of the optic nerve head in zebrafish. Regarding the timing of optic fissure fusion, our data are in line with a recent study [23] showing that the fusion onset is approximately at 35 hpf.

### 2.3. Orientation and Characterization of Pioneer Cells

We described that pioneer cells establish contact between the margins in the upper third of the fissure. We further showed that the pioneer cells are consecutively integrated into the prospective NR. The remodeling of the margins during fusion includes the repositioning/reorientation of the epithelial cells, most importantly the pioneer cells. Thus, we next sought to relate the location of the first cellular contact to the apical-basal orientation of the epithelial cells within the margins. We used the junctional marker n-cadherin in transgenic embryos of the line *tg(ncad:ncad-GFP)* [52], injected with RNA for lyntdTomato into zygotes, for in vivo time-lapse imaging. The apical domain of the cells, facing towards the optic ventricle, was thereby visualized and followed over time (Figure 3A–D’,E, blue line). At contact formation (Figure 3A,A’), the apical domains (Figure 3A, A´,E, arrows) of the future NR, the pioneer cells and the future RPE in the optic fissure region were located ventrally (Figure 3E, dotted line) of the point of contact formation (Figure 3 A,A´,E, asterisks). This implies that the cellular reorientation after contact formation involves only a 90° turn for the pioneer cells to be integrated into the future NR. As fusion proceeded, the apical domains of the future NR and RPE from both margins were connected (Figure 3B–B´ arrows, C–D’, arrow). Our data suggest that the cellular contact was formed by the basolateral domain of the pioneer cells. This likely precedes the apical apposition previously described [22].

At the present time, no specific markers further characterizing the pioneer cells are known. Thus, we investigated markers for NR and RPE with whole mount in situ hybridization (WMISH) to investigate further features of the pioneer cells. The identification of an early marker for the prospective RPE also labeling the ventral domain, which is the last to differentiate, was presumably challenging. Fortunately, dopachrome tautomerase (*dct*) [53] turned out to be a marker expressed in the ventral domain of the future RPE. We found *dct* expression in the lower part of the optic fissure but not reaching the level of pioneer cells (n = 6/7). In one embryo, it was unclear if the level of pioneer cells was reached. This suggests that the pioneer cells within the upper part of the margins are negative for *dct* (Figure 3F,F’).

We used *vsx2* as a classical marker for prospective NR (n = 5). While the prospective NR was largely positive for *vsx2*, the ventral domain of the fissure was found to be negative. However, in the dorsal domain of the optic fissure, the pattern was not as clear. In 2/5 cases, the staining did not reach the level of the pioneer cells. In 1/5 cases, the upper temporal side was negative for *vsx2*, and the upper nasal side positive potentially reaching the pioneer cells. In 2/5 cases, *vsx2* expression was detected on both sides, potentially reaching the level of the pioneer cells (Figure 3G–G1′). We can thus not exclude expression of *vsx2* in pioneer cells.

Taken together, our findings suggest that the pioneer cells are indeed a distinct population of cells. They show a reduced level of *rx2* driven GFP before contact formation and at least in part regain this afterwards They are rather flat in shape, suggestive for RPE [47] but negative for *dct*, and they are potentially positive for *vsx2*.

However, we are not able to conclude to what extent the pioneer cells contribute also to the RPE. Our tracking analyses were mostly hampered by the pronounced flat morphology of the differentiating RPE precursors, which in combination with limitations in spatial and temporal resolution during in vivo time-lapse imaging made reliable cell tracking impossible in many cases. Thus, it will be important to find specific markers for pioneer cells that can be used to perform proper lineage tracing in future analyses.

### 2.4. Pioneer Cells Initiate Optic Fissure Fusion between Two Hyaloid Vessels

Besides the pioneer cell population itself, we identified a tube-like structure over which the pioneer cells extended and eventually established contact (Figure 1C,C’, asterisk). The position, morphology and especially the dynamics of fast cell movements inside this structure, visible during in vivo time-lapse recordings (Appendix A), suggested that this structure was a blood vessel. To substantiate this finding, we performed time-lapse imaging using a transgenic line in which endothelial cells are labelled (*tg(fli1a:eGFP)* [54] (Figure 4). Indeed, we could identify the structure as the hyaloid vein (Figure 4, hv). Prior to the onset of optic fissure fusion, the hyaloid vein entered the optic cup in close proximity to the hyaloid artery at the proximal end of the optic fissure. The fusion process was then initiated in between both vessels, separating them (Figure 4A–C, Appendix A). However, a close interaction only occurred between the fissure margins and the hyaloid vein. Our data suggest that a segment of the hyaloid vessels, located in between the fissure margins, could serve as a scaffold for the margins to bridge the gap (Figure 1, Figure 4C).

As a consequence of optic fissure fusion, the segment of the hyaloid vein over which the pioneer cells made contact was repositioned below the optic cup (Figure 4D–F, Appendix A). This segment thereby became the ventral radial vessel [55], which remained connected to the hyaloid vein distal to the site of fusion (Figure 4G, scheme). Notably, the site of fissure fusion was moving distally, likely pushing the vessel in the same direction (Figure 4D’–F’,G, scheme). This movement of the vessel is in line with a previous observation [22]. Consequently, the distance between the hyaloid vein and the hyaloid artery increased over time (Figure 4G, scheme).

Even though our data suggest a close interaction of the hyaloid vein and the fissure margins during fusion, lack of the early hyaloid vasculature did not prevent, but only delayed basal lamina dissolution [23] and even though other ocular phenotypes were observed resulting from such avascular eyes in the “*cloche*” mutants, no coloboma was reported ([29] and own observations). Since the POM has been demonstrated to play an essential role for fissure fusion [22], likely for basal lamina dissolution, we find it possible that certain POM cells are present in the avascular eyes of the cloche embryos, at least with a temporal delay. This seems likely, considering the inability of the margins to fuse in the absence of POM [22]. While it may be possible that the hyaloid vein indeed serves as a scaffold for pioneer cells in normal eyes, we found no gap between the optic fissure margins in *cloche* embryos (data not shown), which could mean that there is potentially no need for a scaffold in that condition or that other tissue, e.g., POM, is compensating. On the contrary, an abnormally large vessel, resulting from a *lmo2* mutation, was shown to result in coloboma [30]. In this context, the contact could be merely blocked by an enlarged vessel. It is, however, also conceivable that the mutation affected the ECM composition, in turn negatively affecting fissure fusion. The latter, however, seems unlikely since a reduction of hyaloid vein size in the lmo2 mutant was sufficient to rescue the coloboma phenotype [30]. We measured the size of the hyaloid vein during contact formation and during ongoing fusion (Figure 4H–J). The median height and width of the hyaloid vein were 12.7 µm and 11.7 µm, respectively, at the site of initial contact formation and 16.4 µm and 11.9 µm, respectively, during ongoing fusion.

### 2.5. Late BMP Induction Does Not Alter Margin Architecture but Identity and Hampers Basal Lamina Dissolution

BMP and BMP antagonists play important roles during different aspects of eye development, like axis specification and optic cup and optic fissure morphogenesis [1,21,36,37], and also during optic fissure fusion [18]. For optic fissure fusion, BMP signaling must be kept out of the fissure margins, which is accomplished by the local expression of BMP antagonists [18,56]. We had previously induced BMP4 expression experimentally, resulting in coloboma [18]. Notably, the colobomata were displaying a spectrum of morphologies, depending on the timing of BMP induction. Early BMP induction resulted in morphogenetic coloboma [18,21], while late BMP induction resulted in a fusion deficient coloboma [18].

We next investigated how the process of optic fissure fusion is hampered after such late BMP induction. To this end, we made use of the *tg(hsp70l:bmp4)* zebrafish [18] and performed heat shock-mediated induction of BMP4 at a stage of development at which the morphogenesis of the margins was completed (26 hpf), particularly in the proximal domain in which the fusion is initiated in control embryos (Figure 2). We used a BMP responsive reporter (*tg(BRE-AAVmlp:eGFP*) [57] to assess the efficacy of the heat shock protocol and the *tg(hsp70l:bmp4).* The induction of BMP4 resulted in an increase in GFP expression, especially in the ventral domain of the retina including the optic fissure margins, compared to a control embryo (Figure 5A,B).

We next performed in vivo time-lapse imaging to further investigate the reason for the inability of the fissure margins to fuse after BMP induction. We found that the experimental induction of BMP did not negatively affect the architecture of the margins prior to the moment at which the fusion would be initiated under control conditions (Figure 5G–J (n = 13), see C-F as control). The cells lining the optic fissure margins exhibited flat morphology and the strongly reduced level of GFP, driven by the *rx2* cis regulatory element, as seen in control conditions (Figure 5C–J, arrow). Also, the localization of the apical domains did not seem to be affected by the induction of BMP4 (Appendix A). However, even though the margins seemed morphologically unaffected, they were incapable to establish the cellular contact (Figure 5I,J, see Figure 5E–F as control, Appendix A, see Appendix A).

Having shown that the overall architecture of the margins was not affected by BMP induction, we asked whether the identity of the margins could have been altered. To address this, we performed WMISH analyses for a NR marker (*vsx2*), a RPE marker (*dct*), a stalk/marginal marker (*pax2a*), a dorsal optic cup marker [1] (*bambia*) and a ventral optic cup marker (*vax2*) in control and BMP induced embryos.

We found, that BMP induction resulted in an increase of *vsx2* expression in the margins of the optic fissure (Figure 5L–L1′, see K–K1 as control). Notably, *dct* expression was found more everted, outside of the fissure region after BMP induction (Figure 5O–O’, see M as control). This suggests that even though the pioneer cell containing margins seemed morphologically unaffected and even though the *rx2* driven GFP expression ceased in the presumptive pioneer cells, the cell identity has changed within the margins. The analysis of the markers shows that the *vsx2*-positive and the *dct*-positive domains are both everted, compared to controls. Interestingly, the expression of *pax2a*, a marker for the optic stalk and fissure margins, was not changed after BMP induction (Figure 5Q, see P as control). Dorsal and ventral identity markers, however, were affected drastically. *Vax2* was absent from the ventral domain and the *bambia* expression domain in the region of the dorsal optic cup was enlarged after BMP induction (Figure 5S,U, see R and T as controls). The effect of the BMP induction on the dorsal – ventral axis, is in line with previous observations [1,58]. The dorsalization, judged from the *bambia* expression, was not as intense as observed after an earlier BMP induction [1]. The effect on the repression of *vax2* expression, however, was more prominent [1].

We next addressed whether the basal lamina was dissolving after BMP induction. To this end, we performed immunohistochemistry on whole mount processed zebrafish embryos at different developmental stages using an antibody against laminin and compared controls and BMP induced embryos. We observed, that the basal lamina was persisting after BMP induction (Figure 5V–V3, see W–W3 as control).

### 2.6. Late BMP Induction Results in Only Subtle Direct Changes of the POM and Hyaloid Vessel

It was recently shown that the basal lamina dissolution preceding optic fissure fusion depends on the hyaloid vessels or their POM derived progenitors [23]. Thus, we addressed whether BMP induction affects POM and hyaloid vessel development. To visualize POM cells during in vivo time-lapse imaging, we used a *sox10* cis regulatory element driving a membrane localized tdTomato (*tg(sox10:lyntdTomato)*). *Sox10* is an established marker for the neural crest derived cells rather than for other mesenchymal cells of the POM [59,60,61]. We further crossed transgenic fish of this line to *tg(hsp70:bmp4)*. To visualize cell nuclei, we performed mRNA injections of H2B-EGFP into zygotes. To visualize hyaloid vasculature, we used (*tg(fli1a:eGFP)* and crosses to *tg(hsp70:bmp4)*.

We performed heat shocks at 26 hpf and investigated BMP-induced embryos and controls. We recorded 3D stacks over time, starting at 34 hpf and 32 hpf respectively.

We did not observe any consistent gross morphological changes when comparing the control embryos with the BMP-induced embryos at the beginning of our imaging experiments (Figure 6E,M, see A and I as control) 6–8 h after BMP induction. The anlage of the hyaloid vessels could be detected consistently, although the size of the vessels was found to be smaller in a single case. In addition, POM cells were found in controls as well as in BMP induced embryos, in both cases contributing to both the hyaloid artery and the hyaloid vein, which could be found in close proximity to the prospective optic nerve head (Figure 6A,E,I,M). Therefore, we determined that the region of the predicted first cellular contact was properly set up in terms of availability of POM and hyaloid vessels after BMP induction. Importantly, the size of the hyaloid vessels was not found overall enlarged after BMP induction like in the case of an *lmo2* mutation, which resulted in coloboma [30]. Nevertheless, the consecutive interaction of the optic fissure margins and the hyaloid vessel segment failed upon BMP induction (Figure 6E–H,M–P, see A–D and I–L as control, Appendix A, see Appendix A as control). While the margins got in touch with the vessel in control embryos, used it to establish the contact between the margins and thereby separated the hyaloid vein from the hyaloid artery (Figure 6B,C,J,K, 
Appendix A), such margin-vessel interaction could not be observed after BMP induction (Figure 6F,G,N,O, Appendix A). Notably, *sox10*-positive cells and remnants of the hyaloid vessels could be detected in between the fissure margins (Figure 6H,P arrows, see D and L as control).

Overall, our data suggest that the proximal domain of the optic fissure is critical for the onset of optic fissure fusion. Importantly, the fissure does not start to fuse more distally when it fails to fuse in this proximal domain. In a previous study, we identified TGFβ signaling to be important for fissure fusion [18] and subsequently described a TGFβ signaling positive domain of cells entering the optic fissure margins including the future optic nerve head from the optic stalk [1]. Taken together, our data point to the significance of this proximal domain for optic fissure fusion. Based on our own recent findings, regarding the expression of BMP antagonists within the fissure margins [18] and our current investigation, we consider it likely that the margin cells, as well as the POM cells [22,23], substantially contribute to the degradation of the basal lamina. It is also conceivable that there is a crosstalk between the pioneer cells and the POM cells, which is negatively affected by BMP induction. This could explain the persisting remnants of the POM and hyaloid vasculature.

In a next step, the factors mediating the degradation of the basal lamina and thus allowing the pioneer cells to initiate the fusion, need to be addressed. These factors could be derived from either the POM cells or the fissure margin cells. Findings from recent transcriptional analyses likely include potential candidates [18,32,35,62].

### 2.7. BMP-Induced Failure of Optic Fissure Fusion Disrupts Development of Ventral Eye Vasculature

We addressed next whether the BMP induced failure of optic fissure fusion is influencing the consecutive development of the hyaloid vasculature. During normal development, the hyaloid artery, the primary vessel of the optic cup [55,63], can be found within the forming optic nerve head at around 28 hpf. Notably, at the onset of fissure fusion, both the hyaloid artery and the hyaloid vein are positioned in close proximity to the prospective optic nerve head (Figure 4A and Figure 6A,I). These two vessels are only then separated, as the optic fissure margins fuse (Figure 4A–C). When the site of fusion proceeds distally, the hyaloid vein is moving with it towards the periphery (Figure 4D–F, G, scheme).

Thus, when the fusion of the optic fissure margins is hampered after BMP induction, the presumptive driving force behind the separation of the hyaloid artery and vein is absent. We wondered whether BMP4 induction at 26 hpf would therefore lead to both vessels remaining close to each other. To address this, we used the transgenic line *tg(fli1a:eGFP)* [54], labeling blood vessels and crossed this with *tg(hsp70l:bmp4)*. We further added an independent ubiquitous labeling with lyntdTomato, achieved by mRNA injections into zygotes. We analyzed the blood vessels at approximately 48 hpf. Notably, we found the hyaloid vein remaining in the prospective optic nerve head region after BMP induction (Figure 7E, see A as control). In distal regions (Figure 7G, see Figure 7C–D and Appendix A as control), the hyaloid vein, but also components of the superficial ocular vasculature (the ventral radial vessel (VRV) and the ventral part of the annular vessel) were missing. The absence of the VRV can be explained by the hyaloid vein remaining proximal, since the VRV is a segment of the hyaloid vein that is moved out of the optic cup during optic fissure fusion (Figure 4D–F,G, scheme). Furthermore, we assume that the defect of the ventral part of the annular vessel is another consequence of the hyaloid vein remaining proximal. This would be in line with previous findings, showing that the ventral part of the superficial annular vessel is formed by the VRV, while the dorsal part together with the dorsal and nasal radial vessel are formed separately [55]. It is therefore to be expected that ventral parts of the annular vessel are absent if the VRV does not reach distal parts of the eye. The dorsal vessels should however not be affected. Indeed, in our analysis, the dorsal superficial blood vessels of the eye were present. We found both the nasal and dorsal radial vessel (DRV) and the connection between the two (Figure 7G,H), although the DRV was bifurcated. On the temporal side we found an irregular blood vessel (Figure 7 H, asterisk) or a blind ending stump, which we interpret as an aberrant part of the annular vessel that could not connect to the VRV. For a schematic depiction of the vessels see Figure 7 I. A case of optic cup developmental defects impacting vascular development of the eye was also seen in the case of superior ocular coloboma [42]. There, the DRV is formed aberrantly as a result of the superior ocular sulcus, which it uses as guidance, being too deep.

Taken together, our data suggest that the displacement of the hyaloid vein toward the periphery is dependent on the distally progressing site of optic fissure fusion and is required for proper vascularization of the ventral optic cup.

## 3. Materials and Methods

Zebrafish Husbandry. Zebrafish (*Danio rerio*) were kept in closed stocks in accordance with local animal welfare law with the permit 35-9185.64/1.1 (29.06.2016) from the Regierungspräsidium Freiburg. New transgenic fish were generated in accordance with the local welfare law and the permit G-16/160 (09.11.2016) from the Regierungspräsidium Freiburg. The fish facility is under the supervision of the local representative of the animal welfare agency. Fish were maintained in a constant recirculating system at 28 °C on a 12h light, 12h dark cycle. Age of zebrafish embryos was determined according to Kimmel et al. [64]. Fish lines used in this study are bred in AB wild-type background.

Transgenic zebrafish. rx2 reporter: *tg(Ola.rx2:eGFP-caax)* were described and used previously [21]. HS-bmp4: *tg(hsp70:bmp4; myl7:eGFP)* were described and used previously [18]. *(tg(fli1a:eGFP)* [54], tg(*ncad:ncad-GFP)* [52] was a kind gift of Darren Gilmour, and (*tg(BRE-AAVmlp:eGFP)* [57] was a kind gift of Jochen Wittbrodt.

The expression construct for the sox10 reporter *tg(sox10:lyntdTomato)* was assembled in a Gateway reaction, using a Tol2 destination vector pDestTol2pA2 [65] a 5′entry vector containing a sox10 promotor element (kindly provided by Bruce Appel), a middle entry vector containing lyntdTomato, and a 3′entry vector with a polyadenylation site [65]. The construct (10ng µl^−1^) and Tol2 transposase mRNA (7ng µl^−1^) [65] were injected into wildtype zebrafish zygotes. Embryos with strong lyntdTomato expression were selected using a fluorescence stereomicroscope and used as founders for transgenic line generation.

*Transient labelling of zebrafish.* Where indicated, H2BGFP (nuclear localized GFP) (70 ng µl^−1^), lyntdTomato (75–150 ng µl^−1^) was injected into 1–2 cell staged zebrafish embryos enabling four-dimensional imaging of mosaic or ubiquitous labelled zebrafish.

### 3.1. Heat Shock

For all heat shock experiments the *tg(hsp70:bmp4; myl7:GFP)* line was outcrossed to either AB/AB, *tg(sox10:lyntdTomato)*, *(tg(fli1a:eGFP), tg(ncad:ncad-GFP)* or *tg(Ola.rx2:eGFP-caax).* Heat shocks were applied in a heating block at 37 °C for 1 h at 26 hpf. For all heat shock experiments, heat shocked siblings without the transgene (*hsp70:bmp4; myl7:GFP)* were used as controls. These were identified by the absence of GFP expression in the heart (*myl7:GFP)*.

### 3.2. Whole Mount Immunohistochemistry

For whole mount staining embryos were fixed in 4% PFA overnight, washed with PTW, digested with Proteinase K (10µg/mL) for 15 min (32 hpf embryos), 20 min (42 hpf and 43 hpf embryos), 25 min (52 hpf embryos), and 35 min (72 hpf embryos),; incubated in acetone for 20–40 min at –20 °C; and rehydrated in a MeOH series (75/50/25% in PTW). Embryos were washed with PTW and blocked overnight with PBDT (PBS [1×], BSA [1%], DMSO [1%], Triton X-100 [0.1%]) supplemented with 10% normal goat serum (NGS). Primary antibodies anti-Laminin (Laminin Ab1, ThermoFisher, Waltham, MA, USA) 1:100, and where indicated, chicken anti GFP (Invitrogen, ThermoFisher, Waltham, MA USA) 1:500 were accordingly diluted in PBDT supplemented with 10% NGS and incubated for 48 hrs, washed, and incubated with secondary antibody (IgG1 Fab2 Alexa fluor 647, Cell Signaling, Danvers, MA, USA) 1:500 and where indicated DAPI (stock: 4 μg/mL) for 24hpf. After washing with PTW, embryos were embedded in low melting agarose and imaged at a Leica TCS SP8 confocal microscope (Leica, Wetzlar, Germany) with a z-spacing of 3 µm.

### 3.3. Time-Lapse Imaging and Image Processing

Time-lapse imaging was performed on inverted Leica confocal microscopes (TCS SP8, Leica, Wetzlar, Germany) with internal hybrid detectors and a 40x long distance objective (Leica, water immersion). Immersol (Carl Zeiss, Oberkochen, Germany) was used as immersion medium. Embryos at desired developmental stages were embedded in 1% low melting agarose in glass bottom dishes (MatTek, Ashland, MA, USA). Consecutively, they were covered with zebrafish medium, including tricaine for anesthesia. Confocal stacks were recorded every 10 to 12 min with a resolution of 1024 × 1024 and 3 µm z-spacing. Left and right eyes were used and oriented to fit the standard lateral view. Confocal stacks were analyzed using the Fiji software [50]. For denoising of the movies, the PureDenoise plugin [66] was used with four cycle-spins and three frames multiframe.

### 3.4. Quantitative Analysis of the Height of Optic Fissure Fusion

Figure 3A illustrates how the measurement regarding the contact formation was performed. One sphere was drawn to mark the lens-averted border of the optic fissure, and another sphere was placed inside the optic cup to mark the lens facing border. The distance between these borders was measured and defined as 100%. The pioneer cell, establishing the cellular contact, was virtually split in half, and the distance to the ventral border was measured from that point (n = 7). Measurements were performed using Fiji [50].

### 3.5. Analysis of the Proximal-Distal Position of the First Contact (FC)

The location of the future optic nerve head (ONH) was defined to be in the most distal optical plane in which the only visible lumen within the optic cup was the optic fissure. The proximal border of the developing lens (LP) was defined as the most proximal optical plane in which the lens was visible. These two points were used as reference points. The first cellular contact between the margins always occurred in between these two reference points. The distances between the point of first contact and the reference points are shown in µm (n = 9, results are in multiples of three due to stacks being recorded with a z-spacing of 3 µm).

### 3.6. Analysis of the Size of the Hyaloid Vein

First, the height of the hyaloid vein was measured using Fiji [50]. At 50% of the height, the width was measured.

### 3.7. Whole Mount In Situ Hybridization

Whole mount in situ hybridizations were done according to according to Quiring et al. [67] and Heermann et al. [21]. They were stained with FastRed Naphtol (Sigma-Aldrich, St.Louis, Missouri, USA) and DAPI (4 µg/mL) for confocal imaging. For each condition, 8–10 embryos were used, and 5–7 of them were imaged with the confocal microscope. The following probes were newly generated *dct*: Fwd TGTCTAAAGAGTGCTGCCCG, Rev: GGTGCCGTTACACAGAGTCA,

*vsx2*: Fwd CAGGAAAGGATGGGGCTGTT, Rev: ACTCGGGCAGAGGGATAGAG, probes for *pax2a, **bambia* and *vax2* were also previously used [1].

## 4. Conclusions

Coloboma formation can be the result of defects in optic fissure morphogenesis or optic fissure fusion. For a complete understanding of coloboma formation, both processes need further investigation. With our work, we provide more insight into the process of optic fissure fusion. We identified a specific population of cells within the margins, which established the first cellular contact in between the margins. These cells—pioneer cells—do not primarily belong to the pool of NR or RPE progenitors, but after fusion, they integrate into the NR. We further provide insights into the necessity of BMP antagonism in the fissure margins. If BMP signaling is induced within the margins, the identity of the margins is changed, but without morphological changes. Further, the basal lamina is not dissolved, and remnants of POM and hyaloid vessels remain in between the fissure margins, which are unable to fuse. Our data provide an important step toward a better understanding of the fusion process. It will be interesting to further characterize the pioneer cells and also investigate the interaction of these with the POM, which also plays an important role for optic fissure fusion. We finally note the effect the failure of fissure fusion has on the vasculature of the eye.

## Figures and Tables

**Figure 1 ijms-21-02760-f001:**
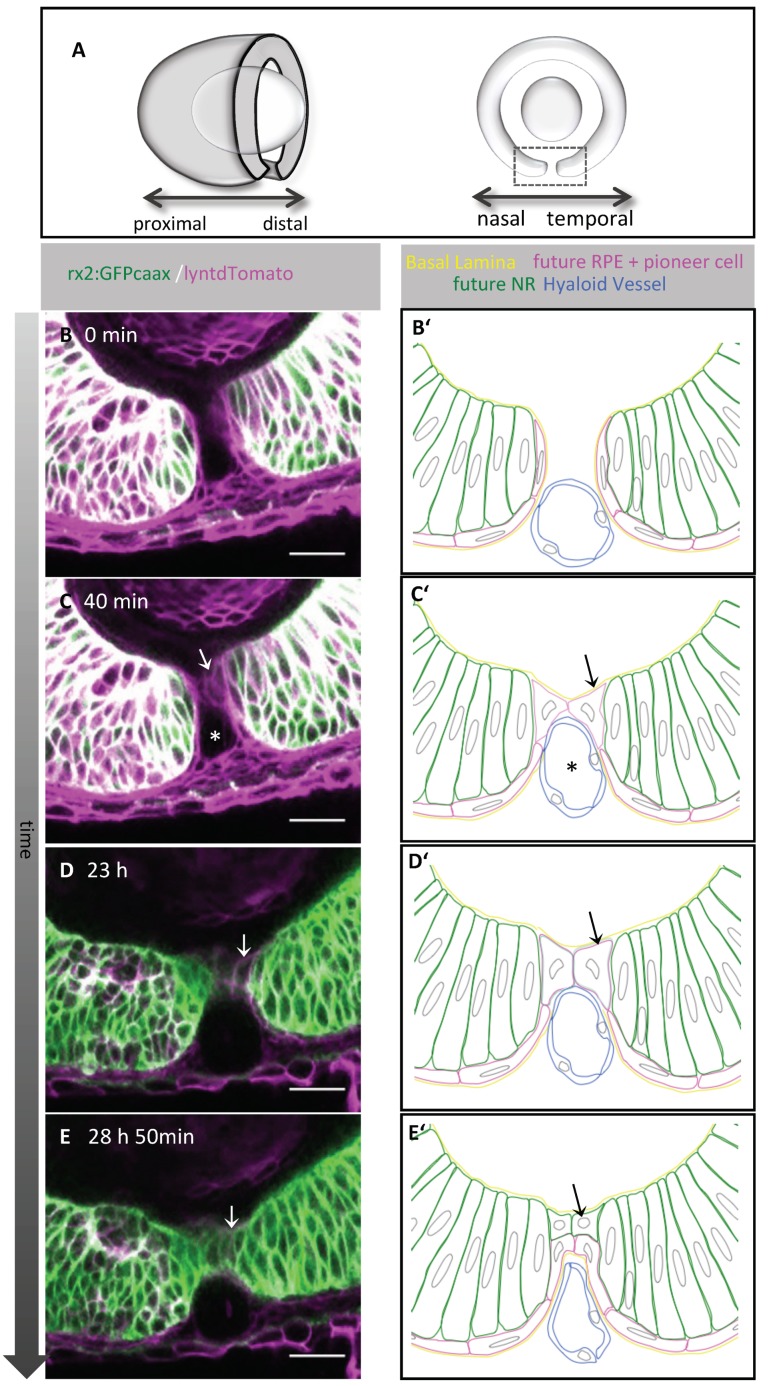
Pioneer cells mediate the cellular contact between the optic fissure margins and consecutively develop into neuroretinal (NR) and likely retinal pigment epithelial (RPE) cells. (**A**) Scheme of a transverse view on the optic cup and a scheme of a sagittal section through the optic cup, the box represents the region depicted in (**B**–**E’**). (**B**–**E**) Single z-plane of a time-lapse 3D confocal imaging stack during fusion of the optic fissure. *tg(rx2:eGFPcaax)* labels neuroretinal precursors (green), zygotic microinjection of lyntdTomato mRNA labels all plasma membranes (magenta). (**B**–**E´**) Schematic representation of fissure fusion. (**B**–**B´**) Optic fissure margins before establishment of the contact. Magenta labeled, but *rx2* reduced/almost negative cells lining both nasal and temporal margins. (**C**,**C´**) Pioneer cells establish the contact between the optic fissure margins (arrow). The lumen of the hyaloid vein is seen ventral of the pioneer cells (asterisk). (**D**,**D´**) Cells, which established the contact, reactivate *rx2* expression (arrow). (**E**,**E´**) Optic fissure after fusion, the *rx2* reporter is active in the lens-facing pioneer cells (arrow), onset of imaging: 37hpf, scale bars 25µm, lateral view, nasal to the left (n = 11).

**Figure 2 ijms-21-02760-f002:**
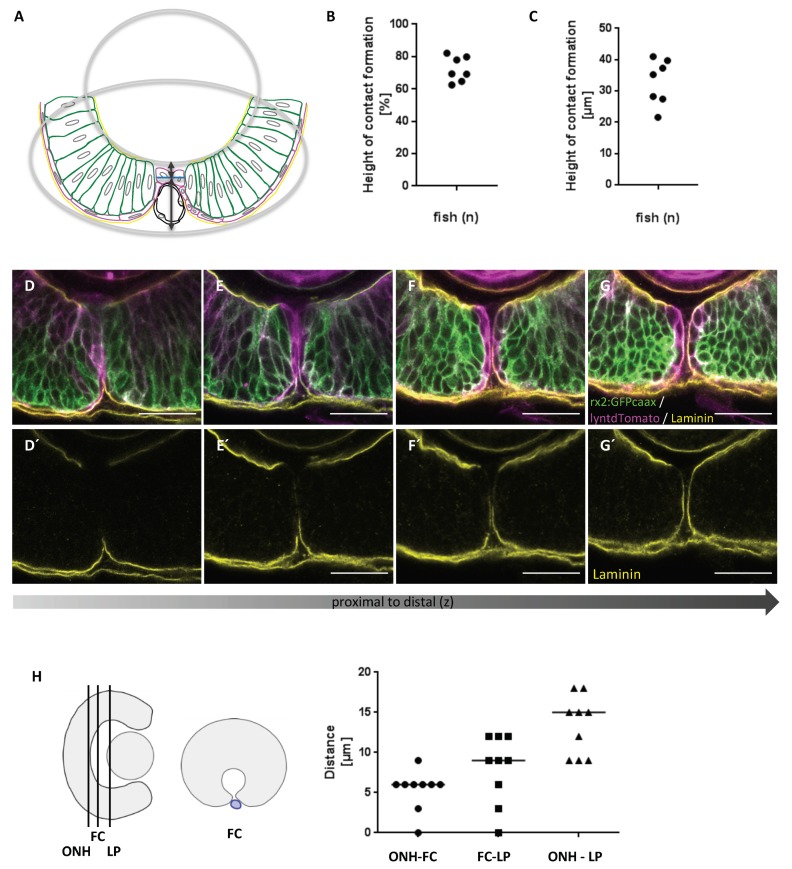
Pioneer cells establish contact between the optic fissure margins in the upper third of the fissure, starting close to the future optic nerve head. (**A**–**C**) Measurement of the height of contact formation between the fissure margins. (**A**) Scheme illustrating the method of measurement. A detailed description is provided in the Materials and Methods section. (**B**,**C**) Position of the point of contact between the margins in (**B**) relation to the height of the fissure and (**C**) absolute distance from the ventral border of the optic cup (n = 7). (**D**–**G´**) Degradation of the basal lamina during contact formation from proximal to distal (left to right). (**D**,**D´**) Laminin is already degraded in the more proximal part of the optic fissure. (**E**–**E´**) Degradation of laminin is extending to the lower part of the fissure to allow fusion of the margins. (**F**,**F´**) Degradation of laminin starts in the upper third of the optic fissure where the pioneer cells are first apposed. (**G**,**G´**) The basal lamina is still intact in a more distal domain where the fissure margins are not yet closely apposed. Four optical sections with a z-spacing of 12µm (**D**–**E´**) and 6 µm (**E**–**G´**) zebrafish eye during optic fissure closure fixed at 43 hpf. (**D**–**G**) *tg(rx2:eGFPcaax)* labels neuroretinal precursors (green), zygotic microinjection of lyntdTomato mRNA labels all plasma membranes (magenta). Embryos were stained with anti-Laminin1 (yellow) and anti-GFP (green), lyntdTomato survived the processing procedure (magenta). (**D´**–**G´**) respective single channel for laminin. Scale bars 25µm, sagittal view, nasal to the left (n = 10). (**H**) Measurement of the point of first contact between the margins (FC) in relation to the proximal-distal axis of the developing optic. The measurements were related to reference points, being the future optic nerve head (ONH) and the proximal end of the developing lens (LP), depicted in the scheme (n = 9).

**Figure 3 ijms-21-02760-f003:**
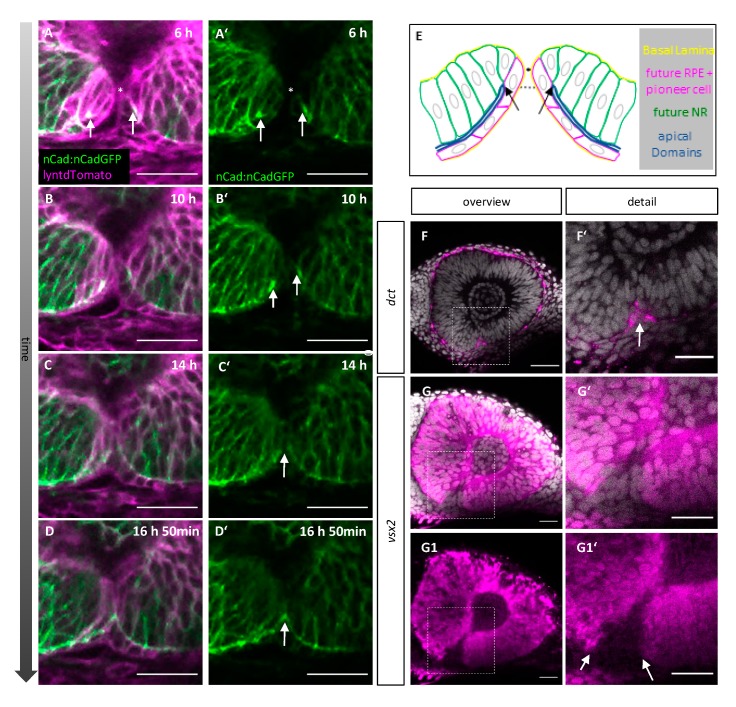
(**A**–**E**) Pioneer cells establish contact above the apical-apical border between the outer and inner layer of the optic cup. (**A**–**D**) Single z-plane of a time-lapse 3D confocal imaging stack during fusion of the optic fissure. *tg(ncad:ncad-GFP)* labels apical adherens junctions between neighbouring cells (green), zygotic microinjection of lyntdTomato mRNA labels all plasma membranes (magenta). Onset of imaging: 32hpf, scale bars 25µm, sagittal view, nasal to the left (n = 3). (**A’**–**D’**) single green channel of (**A**–**D**). (**A**–**B**´) Contact between the optic fissure margins (asterisk) is established above the border between both retinal layers (arrows). (**C**–**D’**) As fusion proceeds, the apical–apical border becomes continuous between both margins (arrow). (**E**) Scheme representing the state in (**A**). (**F**–**G**) In situ hybridizations at 37 hpf for *dct* (F,F’) and *vsx2* (**G**–**G1**′), DAPI in grey. (**F**–**F’**) *dct* is expressed in RPE cells around the eye and in flat cells within the lower part of the optic fissure margins (**F**´arrow, **F**’ detail from **F**), scale bars 25µm. (**G**–**G1**′) *vsx2* is expressed in the NR but not in the optic fissure margins (arrows). (**G’**) detail of (**G**), (**G1**,**G1**′) isolated magenta channels of (**G**,**G’**). Scale bars 25µm (n ≥ 5).

**Figure 4 ijms-21-02760-f004:**
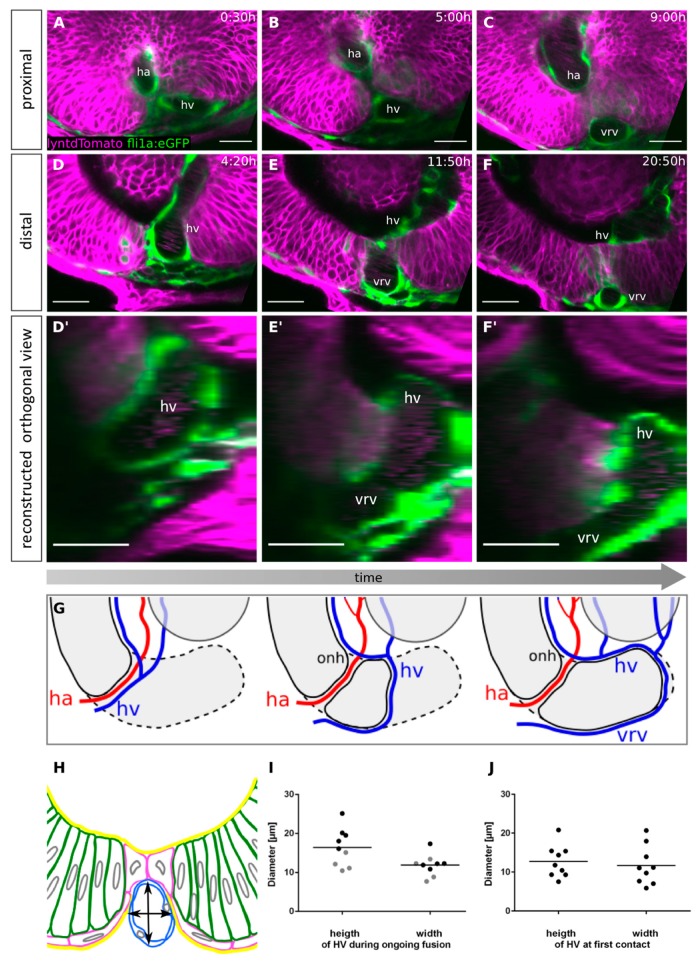
Optic fissure fusion occurs between hyaloid artery and vein and displaces the hyaloid vein towards the distal optic cup. (**A**–**C**) Single z-plane of a time-lapse 3D confocal imaging stack in which the first contact of the optic fissure margins occurs. Sagittal view, nasal to the left. (**D**–**F**) Single z-plane of a time-lapse 3D confocal imaging stack in which the hyaloid vein is moving distally due to optic fissure fusion. (**D**) The hyaloid vein is located in the optic fissure. (**E**) The optic fissure margins bridge over the hyaloid vein. (**F**) Fusion of the fissure margins is complete, and the hyaloid vein is found within the optic cup. The outer part of the vein is now the ventral radial vessel. Sagittal view, nasal to the left. (**D’**–**F’**) Reconstructed transversal views of the time points depicted in D–F, proximal to the left. Onset of imaging: 28hpf, scale bars 25µm, (n = 4). (**G**) Scheme illustrating the hyaloid vasculature before, during, and after optic fissure fusion in a transversal view, proximal to the left. ha: hyaloid artery, hv: hyaloid vein, onh: optic nerve head, vrv: ventral radial vessel. Red indicates arteries, blue indicates veins. Dashed lines mark the boundaries of retinal tissue in other planes. (**H**–**J**) Measurement of the size of the HV during contact formation. (H) Scheme illustrating the method of measurement. (**I**) Height and width of the HV during ongoing fusion, (n = 9), grey data points where measured in the plane of LP (compare Figure 2G), black data points distal of LP. (**G**) Height and width of the HV at first contact formation, (n = 9).

**Figure 5 ijms-21-02760-f005:**
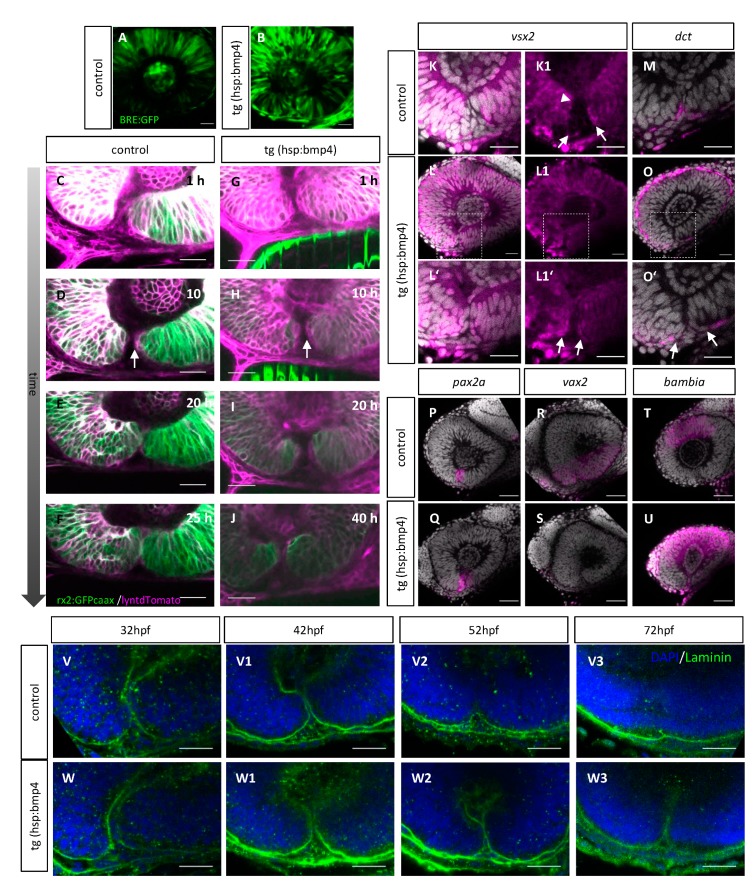
Optic fissure margins do not fuse following *bmp4* induction. (**A**–**B**) Confocal images of (*tg(BRE-AAVmlp:eGFP*) embryonic eyes (41 hpf) after heat shock at 26 hpf. (**A**) BMP signaling is active in the dorsal part of the eyes of control embryos. (**B**) BMP signaling is active in all parts of the eyes of *tg(hsp70l:bmp4)* embryos. (**C**–**J**) Single z-plane of a time-lapse 3D confocal imaging stack including the optic fissure region of (**G**–**J**) a *tg(hsp70l:bmp4)* embryo ( n = 13) and (**C**–**F**) a control sibling (n = 4) starting at 30 hpf after heat shock at 26 hpf. *tg(rx2:eGFPcaax)* labels neuroretinal precursors (green), zygotic microinjection of lyntdTomato mRNA labels all plasma membranes (magenta). The bright green stripes in **G**–**H** originate from the transgenesis marker *myl7:eGFP* in the heart. (**D**,**H**–**J**) The margins of the optic fissure are lined with flat magenta positive cells (arrow). As development proceeds, the fissure margins get closer to each other but do not fuse. (**C**–**F**) Optic fissure fusion is occurring normally in the control. (**K**–**O**) In situ hybridizations at 37 hpf for (**K**,**L**) *vsx2* and (**M**,**O**) *dct*. (**K**) *vsx2* partially expressed in the upper third of the fissure margins of wild type embryos (K1 arrowhead) but not in the lower part of optic fissure margins (K1 arrows). (**L**) *vsx2* is expressed in the NR and optic fissure margins (**L1´**arrows) after *bmp4* induction at 26 hpf, (**L’**) detail of (**L**). (**K1**–**L1′**) isolated magenta channels of (**K**–**L1**). (M) Expression of dct extends into the fissure margins of wild type embryos. (**O**) dct expression is not found within the optic fissure margins after bmp4 induction (arrows). (**O’**) detail of (**O**), scale bars 25µm, (n ≥ 5 for all). (**P**–**U**) In situ hybridizations at 37 hpf for (P,Q) *pax2a*, (R,S) *vax2*, and (**T**,**U**) *bambia*. (**P**) *pax2a* is expressed in the optic fissure region of wild type embryos. (**Q**) Expression of *pax2a* is not changed after bmp4 induction. (**R**) *vax2* is expressed in the ventral half of the optic cup of wild type embryos. (**S**) *vax2* was not detected in the optic cup after timed induction of *bmp4* at 26hpf. (**T**) *bambia* is expressed in the dorsal part of the optic cup of wild type embryos. (**D**) In *bmp4*-induced embryos, *bambia* is present in the dorsal half of the eye. scale bars 50µm, (n ≥ 5 for all). (**V**–**W**) Whole mount anti-Laminin staining of (**W**–**W3**) *tg(hsp70l:bmp4)* embryos and (**V**–**V3**) control siblings at different developmental stages (32, 42, 52, 72 hpf) after heat shock at 26 hpf (n ≥ 5 for each condition). (**V**–**V3**) The the basal lamina is orderly dissolving in the control siblings. (**W**–**W3**) Optic fissure margins are in close contact to each other, the basal lamina is remaining intact in the heat shocked *tg(hsp70l:bmp4)* embryos. All images present sagittal views, nasal to the left.

**Figure 6 ijms-21-02760-f006:**
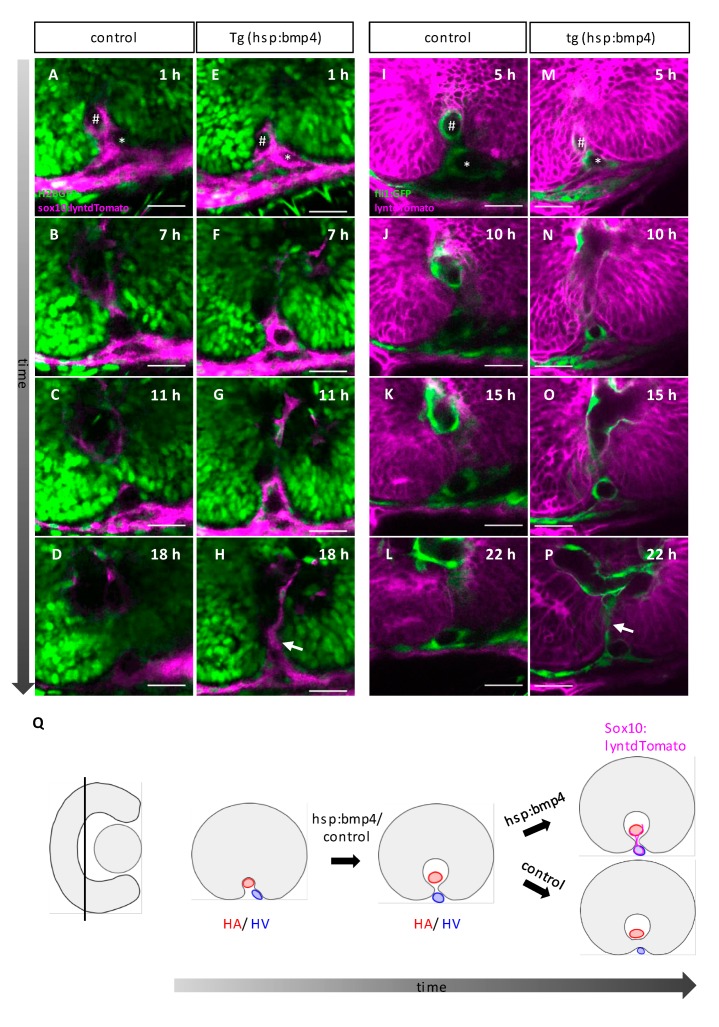
Optic fissure margins are not able to fuse between HV and HA after *bmp4* induction. (**A**–**H**) Single z-plane of a time-lapse 3D confocal imaging stack including the proximal optic fissure region of (**E**–**H**) a *tg(hsp70l:bmp4)* embryo and (**A**–**D**) a control sibling starting at 34 hpf after heat shock at 26 hpf. *tg(sox10:lyntdTomato)* labels membranes of POM cells and their progeny (magenta), zygotic microinjection of H2B-eGFP mRNA labels all nuclei (green). (**I**–**P**) Single z-plane of a time-lapse 3D confocal imaging stack including the proximal optic fissure region of (**M**–**P**) a *tg(hsp70l:bmp4)* embryo and (**I**–**L**) a control sibling starting at 32 hpf after heat shock at 26 hpf. *tg(fli1a:eGFP)* labels endothelial cells (green), zygotic microinjection of lyntdTomato mRNA labels all plasma membranes (magenta). (**A,E,I,M**) The distribution of POM cells/ the formation of the BV is normal in *tg(hsp70l:bmp4)* embryos. The lumen of the hyaloid artery (**A**,**E**,**I**,**M** hash) and vein (**A**,**E**,**I**,**M** asterisk) can be seen in the optic fissure region. (B–D,F–G,J–L,N–P) POM cells/hyaloid vessels remain within the optic fissure in *tg(hsp70l:bmp4)* embryos (**H,P** arrow) (n = 4, n = 3 respectively), while the fissure is closed control embryos (n = 2/n = 2). Scale bars 25µm, sagittal view, nasal to the left. (**Q**) Scheme of the findings from (**A**–**P**). Hyaloid artery and vein are separated by the onset of fissure fusion. In control embryos and *sox10*/*fli* positive cells disappear from the fissure. In *tg(hsp70l:bmp4)* embryos, they remain between the fissure margins. It is unclear whether this is cause or consequence of the failure of optic fissure fusion.

**Figure 7 ijms-21-02760-f007:**
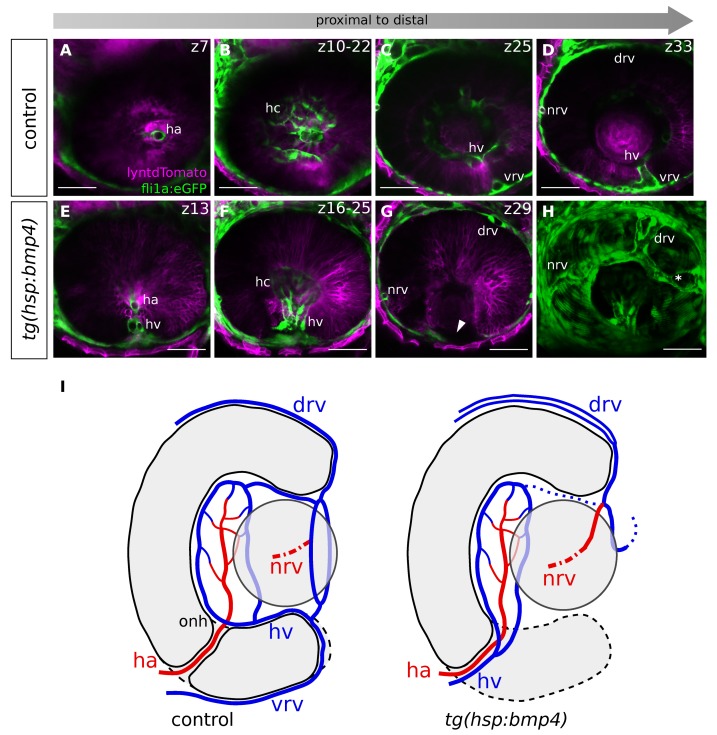
Impaired optic fissure fusion in *tg(hsp70l:bmp4)* embryos results in the hyaloid vein remaining proximal, and incomplete formation of the superficial annular vessel. (**A**–**D**) Confocal imaging of a control embryonic eye with blood vessels at approx. 48hpf after heat shock at 26hpf (n = 7), showing single z-planes or maximum projections. Sagittal view, nasal to the left. (**A**) Optic nerve head region with hyaloid artery. (B) Maximum projection showing the hyaloid capillaries. (**C,D**) Distal ocular regions with hyaloid vein, ventral radial vessel, and the connection between the two. (E–H) Confocal imaging of a *tg(hsp70l:bmp4)* embryonic eye with blood vessels at approximately 48 hpf after heat shock at 26 hpf (n = 12), showing single z-planes or maximum projections. Sagittal view, nasal to the left. (**E**) Optic nerve head region with hyaloid artery and abnormal hyaloid vein. (**F**) Maximum projection showing the hyaloid capillaries. (**G**) Distal ocular region. The persisting optic fissure is devoid of vessels (arrowhead). (**H**) Full maximum projection of the *tg(hsp70l:bmp4)* ocular vessels. The nasal and dorsal radial vessels are present, although the latter is abnormally bifurcated. There is an additional superficial vessel on the temporal side (asterisk). The ventral radial vessel and the ventral parts of the annular vessel are missing. Scale bars 50µm, z-spacing 3µm. (**I**) Scheme illustrating the vascular system of control and *bmp4*-induced embryos in a transverse view. Note that the nasal radial vessel lies in a different plane (indicated by a dashed line). Dotted lines mark inconsistent vessels. Red lines indicate arteries, blue lines veins. Ha: hyaloid artery, hv: hyaloid vein, hc: hyaloid capillaries, nrv: nasal radial vessel, drv: dorsal radial vessel, vrv: ventral radial vessel, onh: optic nerve head.

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
