# Peer review of "In Vivo Analysis of Optic Fissure Fusion in Zebrafish: Pioneer Cells, Basal Lamina, Hyaloid Vessels, and How Fissure Fusion is Affected by BMP"

_ijms, 2020, doi:10.3390/ijms21082760_

Round 1

Reviewer 1 Report

   Since the last review the authors have attempted to correct deficiencies and add new data to the manuscript. While some aspects have been addressed, including better reporting of scientific rigor, data representation and more detailed methods, several significant concerns from past reviews remain. In particular, I remain unconvinced that the authors classification of pioneer cells as non NR is sufficiently supported by their data. New experiments attempt to shed light on this issue, but in their current state appear to muddy the waters further.

   I feel the manuscript would better serve the community as a descriptive in vivo examination of early OF fusion events without claims of pioneer cell identity. In vivo analysis of this complex event is a needed addition to the field and this work represents a nice start in that direction. The addition of BMP regulation is also informative, although not overly novel. Whether just the descriptive nature of this work would be of sufficient impact for the journal would be at the discretion of the editor. Pursuit of the molecular identity of pioneer cells, a very important question, should be an independent project, one that will require significantly more investigation and rigor.

Major Comments

The author’s explanation of fixation artifacts for why pioneer cells are not observed in previous examination of fissue fusion is not convincing. I am unaware of fixation altering OF basement membrane integrity to the point that the physical presence of pioneer cells would be lost upon fixation and appear as a smooth and uninterrupted basement membrane. Instead, I suggest the authors should visualize laminin, membrane and rx2:GFP simultaneously. This experiment will consolidate the interaction of pioneer cells, the basement membrane and fusion mechanics and would be a very important and significant addition to the story.

The authors continue to imply that rx2:GFP is a direct readout of rx2 expression and therefore identity of these cells, (line 147). I remain firm on the requirement of examining endogenous rx2 expression in order to make this claim. GFP signal is a readout of GFP protein stability, and GFP is highly stable. It would make more sense if these were newly born cells that have yet to express GFP via the rx2 promoter, rather than as suggested in the manuscript cells that somehow turn of rx2 driven GFP expression and also remove the GFP protein. To that point, the authors do not show any tracking data to support that these cells were, or were not, GFP+ at some point before they are examined in Figure 1.

   -Second, when looking at supplemental figure 1 depicting the individual channels it appears to me that the pioneer cells are in fact GFP+. These cells appear somewhat out of focus, which is also apparent in the dTomato channel. Based on this data I am not convinced that these pioneer cells are in fact GFP-. Quality of the supplemental figure images is also very poor.

Overall, I feel the authors should not make claims that the pioneer cells are somehow non-NR, the supportive evidence for such a claim is lacking. Observing their behavior in real time is important and informative but classifying their molecular identity convincingly may be out of the scope of this study.

Addition of new data using Ncad-GFP is a good idea, however, the data provided is not overly convincing. Movie 2 is of very low quality and it is very difficult to make out the points the authors try to address in the manuscript. I see cell movement, but would not feel confident about conclusions of specific events. From examining the data I am not able to ascertain how the authors made the claims of cellular reorientation. Also, why are the pioneer cell contacts being observed in the ventral regions of the fissure, where in Figure 1 the cells were making contacts in the dorsal regions? There is no attempt made at quantification of the observations. Also, does BMP activation affect the polarity of pioneer cells? Lastly, authors mention that their findings are at odds with previous published results, but do not comment on this further. I am not sure this data adds important detail to the story, and in its current form it actually detracts from it.

Addition of vsx2 expression could be useful, however, the suggestion that the vsx2 negative cells are pioneers is not consistent with their location in the fissure, since the more dorsal regions are already opposed and show strong vsx2 expression. The only negative cells are in the ventral region of the fissure and could very well represent RPE or POM/vasculature cells (Fig 5K,K1). Furthermore, the authors go on to state that dct (RPE) expression occurs in this particular region, and if you compare dct expression to the cells negative for vsx2 they look like the same population. In addition, the figures lack an indication of where the inset (detail) region is from. Overall, this is new data is very subjective to interpretation and lacks any quantitation in either reproducibility between experiments or timing during development (when does vsx2 turn off in these cells, do they every express vsx2, do they express vsx2 after fusion?).

  Comparison of vsx2 expression vs BMP induction again attempts to claim a difference in vsx2 expression in the fissure margins, however, the images in the figures are not convincing. Figure 5 K vs L shows optical planes that are not exactly comparible, control images display a significantly smaller lens region and lack the level of apposition of the retinal lobes observed in the BMP treated sample. Lastly, I do not see a clear difference in the expression of vsx2 amongst the treatments at the resolution used. Is the arrow in panel K1 suggesting that the entire lobe is missing vsx2 expression (this was not observed in figure 3)? The images are of low quality and therefore difficult to determine presence or absence of expression in specific cells.  In order to be valid, all the vsx2 expression data needs to be quantified, simple qualitative observation comparison is too subjective.

Figure 7: The authors need to show comparable images for their treatments. Fig 7H lacks an accompanying control image (max intensity projection of the control embryo)

Minor point.

The authors indicate that their and other work have found that fusion initiates at ~37hpf, yet their laminin data, Figure 5, suggests that up to 42hpf the basement membrane is intact. Does this indicate that pioneer cell interaction, and therefore initial fusion proceeds without basement membrane removal? The authors should talk about this discrepancy between their and published work.  

Author Response

Reviewer 1

 Since the last review the authors have attempted to correct deficiencies and add new data to the manuscript. While some aspects have been addressed, including better reporting of scientific rigor, data representation and more detailed methods, several significant concerns from past reviews remain. In particular, I remain unconvinced that the authors classification of pioneer cells as non NR is sufficiently supported by their data. New experiments attempt to shed light on this issue, but in their current state appear to muddy the waters further.

We want to thank the referee for the overall detailed comments and especially for again taking the time and effort to review our work. Please see below our response to the constructive comments of the referees.

 I feel the manuscript would better serve the community as a descriptive in vivo examination of early OF fusion events without claims of pioneer cell identity. In vivo analysis of this complex event is a needed addition to the field and this work represents a nice start in that direction. The addition of BMP regulation is also informative, although not overly novel. Whether just the descriptive nature of this work would be of sufficient impact for the journal would be at the discretion of the editor. Pursuit of the molecular identity of pioneer cells, a very important question, should be an independent project, one that will require significantly more investigation and rigor.

We thank the referee for the assessment. It is, however, difficult to comprehend the referees intention. We sense that the referee specifically doubts that pioneer cells are not primarily belonging to the pool of RPE or NR precursors. Yet, to us it seems that the referee agrees with the fact that pioneer cells exist. We identified this specific population of cells for the first time and are aware of the impact. As we pointed out in the initial cover letter and the previous rebuttal letter, our pre-published data has already been used and cited by others.

We agree with the referee that additional investigation is needed to characterize the pioneer cells further. Nevertheless, we provide data which from our point of view is more than sufficient for the first description of the newly identified population of cells which is initiating optic fissure fusion. In some aspects we noticed that we had to further improve the presentation of our data enabling the reader to better understand it. We further considered especially the referees comments regarding the vsx2 expression. Please see our revised work and our comments to the points raised.

In principle, we tend to hold on to the way we presented our data including the aspect of pioneer cells and their first description. Please see our answers to the comments point by point below. We are confident that the referee will find this adequate.

Major Comments

The author’s explanation of fixation artifacts for why pioneer cells are not observed in previous examination of fissue fusion is not convincing.

Here, the referee points to our attempt to find an explanation, why pioneer cells were not found and described in former studies performed by others, when the basal lamina was used as a readout of fissure fusion.  This point was raised during the first round of revisions.

We argued with the following three points and not just termed the previous data “fixation artifacts”:

  • differences in tissue treatment (fixation, processing for immunohistochemistry)
  • use of different labels for structures (membranes vs. nuclear)
  • temporal dynamics (benefit to visualize in vivo and importance for us for the identification in the first place)

Importantly, we pointed out to a different reviewer that in a previous study in vivo and fixed tissues were used side by side resulting in a different appearance of the fissure margins. We include our answer below:

“The referee is likely pointing towards the Figure 1 of Gestri et al., 2018. The differences in

the appearance of the fissure margins are likely due to the differences of tissue treatment and

can be seen even within that very figure. In B-E e.g. fixed tissue is depicted, whereas in I

living tissue is shown….”

To further investigate ourself how the architecture of the optic fissure is changed when the embryos are processed for immunohistochemistry we made a direct comparison between in vivo appearance and appearance after tissue processing.  Please see the new Figure 2 Supplement and the respective legend.

We included an experiment, where we did in vivo imaging of the optic fissure at 43 hpf, than we fixed the embryos, subjected them to immunohistochemistry and imaged again (siblings). What we find is that the hyaloid vein between the fissure margins shrinks upon the treatment and the margins are closer to each other. Thus it is likely that in fixed tissue we cannot see the pioneer cells extend from both sides over the blood vessel, because they are already close to each other (please see new Figure 2 Suppelement, the respective legends the small changes in the  respective Materials and Methods section and the brief notion in the main text in line 240 ff. Please also see the comments below.

Overall, we were also a bit surprised that we found the pioneer cells for the first time. One reason might be that we found a way to label the prospective “RPE” cells especially in combination with the label of the rx2 driven GFP in combination with in vivo imaging. Using only the rx2 driven GFP, it is impossible to visualize the fusion dynamics properly (own observations). Only the use of an independent ubiquitous label enabled us to observe the pioneer cells and their activity. In addition, to our knowledge, there are no RPE precursor reporter fish available.  This is, however, only an explanation we found for us. We of course cannot be sure exactly why others did not find and describe the pioneer cells previously.

I am unaware of fixation altering OF basement membrane integrity to the point that the physical presence of pioneer cells would be lost upon fixation and appear as a smooth and uninterrupted basement membrane.

Instead, I suggest the authors should visualize laminin, membrane and rx2:GFP simultaneously. This experiment will consolidate the interaction of pioneer cells, the basement membrane and fusion mechanics and would be a very important and significant addition to the story.

We performed a triple labeling as the referee suggested to visualize the basal lamina, cell membranes and also the rx2 driven GFP. To this end we used zygotes of tg(rx2:GFPcaax) and injected RNA for lyntdTomato. At 43hpf, we fixed and subjected the embryos to immunohistochemistry using an anti-laminin antibody and anti-GFP antibody. Fortunately, the fluorescent protein lyntdTomato at least partially survived the procedure.

Please see the revised Figure 2 and the new Figure 2 Supplement 1 and 2, the respective legends the small changes in the  respective Materials and Methods section and the brief notion in the main text in line 240 ff. “Notably, the appearance of the site of fissure fusion is different in samples subjected for immunohistochemistry (Figure 2 D-G’, Figure 2 Supplement A-G´´), with the fissure margins being in closer contact overall, likely due to a shrunken hyaloid vessel which increases the space between the margins in vivo (compare to Figure 1 C and C’, asterisk, Figure 2 supplement A´´-H , showing a comparison). ”

The authors continue to imply that rx2:GFP is a direct readout of rx2 expression and therefore identity of these cells, (line 147). I remain firm on the requirement of examining endogenous rx2 expression in order to make this claim.

We thank the referee for this notion and apologize for not having put the statement accurate enough. We changed the sentence in line 203 accordingly. It now reads: “Our data suggest an additional event during which rx2 expression, followed here by using an rx2 cis regulatory element driving GFP expression, is dynamic, including repression followed by a reactivation.

GFP signal is a readout of GFP protein stability, and GFP is highly stable. It would make more sense if these were newly born cells that have yet to express GFP via the rx2 promoter, rather than as suggested in the manuscript cells that somehow turn of rx2 driven GFP expression and also remove the GFP protein. To that point, the authors do not show any tracking data to support that these cells were, or were not, GFP+ at some point before they are examined in Figure 1.

Here, the referee is raising an important point. We are aware of our rather indirect analysis. However, considering the high stability of GFP, it is even more important to note that the GFP is strongly reduced/ almost or totally absent from the pioneer cells. This is essentially what we claim and also show. Furthermore, newborn cells derived from retinal progenitor cells would also not be GFP negative, as they would inherit the membrane-bound protein from their progenitor. We are sorry if we did not made this aspect clear enough. For further clarification please visit Eckert et al., 2019 https://rs.figshare.com/articles/Supplemental_Movie_16_avi_from_Morphogenesis_and_axis_specification_occur_in_parallel_during_optic_cup_and_optic_fissure_formation_differentially_modulated_by_BMP_and_Wnt/7642547

and Heermann et al., 2015 suppl. Movie file 8. We are confident that this provides further insights regarding the earlier expression of rx2 and clarifies this aspect.

   -Second, when looking at supplemental figure 1 depicting the individual channels it appears to me that the pioneer cells are in fact GFP+. These cells appear somewhat out of focus, which is also apparent in the dTomato channel. Based on this data I am not convinced that these pioneer cells are in fact GFP-. Quality of the supplemental figure images is also very poor.

As pointed out above, we claim and show that the GFP is strongly reduced/ almost or totally absent from the pioneer cells. Regarding the reduction of rx2 driven GFP in the pioneer cells, we took extra care. We analyzed 11 datasets/embryos from 3 experiments to come to our conclusion. The data presented is confocal. We do not think that the pioneer cells seem out of focus.

Potentially the quality of the images provided was harmed during the upload. We uploaded the supplemental Figure again. Besides, we added 3 asterisks in A and A’, for enabling easier comparison of the domains and cells in the separated channels. Please see the revised supplemental figure 1 and the corresponding figure legend.

Overall, I feel the authors should not make claims that the pioneer cells are somehow non-NR, the supportive evidence for such a claim is lacking. Observing their behavior in real time is important and informative but classifying their molecular identity convincingly may be out of the scope of this study.

As pointed out in the beginning we agree with the referee that additional investigation is needed to characterize the pioneer cells further. We do not state that we provide a full description of the pioneer cells and we furthermore agree that addressing their molecular identity is beyond the scope of the current study. Nevertheless, we cannot ignore our data and state that the pioneer cells belong to the NR precursor pool primarily. The cells do show features of RPE precursors, they were classified as RPE precursors before (Hero et al., 1990). After contact formation, however, cells do integrate into the pool of NR precursors, as we show and state. From our point of view, it would be a mistake to consider the pioneer cells merely as NR precursors.

Addition of new data using Ncad-GFP is a good idea, however, the data provided is not overly convincing. Movie 2 is of very low quality and it is very difficult to make out the points the authors try to address in the manuscript. I see cell movement, but would not feel confident about conclusions of specific events. From examining the data I am not able to ascertain how the authors made the claims of cellular reorientation.

Also, why are the pioneer cell contacts being observed in the ventral regions of the fissure, where in Figure 1 the cells were making contacts in the dorsal regions?

There is no attempt made at quantification of the observations. Also, does BMP activation affect the polarity of pioneer cells?

Lastly, authors mention that their findings are at odds with previous published results, but do not comment on this further.

We thank the referee for pointing this out. We now carefully revised our statement. Please see the revised version of the manuscript. Line 342 f.

I am not sure this data adds important detail to the story, and in its current form it actually detracts from it.

It was suggested to us in the first round of revision to take into account the cell polarity. We followed the suggestion and addressed as written in 2.3 of our manuscript: “Thus, we next sought to relate the location of the first cellular contact to the apical-basal orientation of the epithelial cells within the margins.”

A little further we note:

“At contact formation (Figure 3A and A’), the apical domains (Figure 3E, scheme, blue line) of the future NR, the pioneer cells and the future RPE in the optic fissure region were located ventral of the point of contact (Figure 3A-D’).”

We think that this notion of ours could have resulted in the misunderstanding. We did not state that the contact is established in the ventral region. On the contrary, the contact formation is in the upper third, however, the apical domains of the cells, this is what we addressed with the Ncad- GFP data, is located further ventral. This is depicted in the scheme of Figure 3. We added further markings to the scheme to point to the apical domains. We further added notes to markings to the main text in section 2.3. Please see the revised section in the track changes modus.We are confident that this and our explanation adds more clarity.

The referee is right, we made this a qualitative description and related the point of contact formation to the level of the apical domains.

The referee further asked whether BMP induction is altering the polarity of the pioneer cells. We performed additional experiments and set up crosses of of tg(Ncad:NcadGFP) and tg(hsp:bmp4).

Briefly, in control and bmp induced embryos the apical domains marked by the Ncad-GFP are reaching into the optic fissure (arrows), to a comparable degree. Please see the new Figure 5 Supplement and the respective legend. We included a notion in Line 641 f.

Addition of vsx2 expression could be useful, however, the suggestion that the vsx2 negative cells are pioneers is not consistent with their location in the fissure, since the more dorsal regions are already opposed and show strong vsx2 expression.

The only negative cells are in the ventral region of the fissure and could very well represent RPE or POM/vasculature cells (Fig 5K,K1). Furthermore, the authors go on to state that dct (RPE) expression occurs in this particular region, and if you compare dct expression to the cells negative for vsx2 they look like the same population.

In addition, the figures lack an indication of where the inset (detail) region is from.

We thank the referee for this notion. We now inserted a box to indicate where the detail was taken from.

Overall, this is new data is very subjective to interpretation and lacks any quantitation in either reproducibility between experiments or timing during development (when does vsx2 turn off in these cells, do they every express vsx2, do they express vsx2 after fusion?).

We could not provide data concerning the timing of vsx2 expression. We consider it likely that after contact formation it will be positive in the prospective NR in the former fissure domain.

Nevertheless, we carefully revisited our vsx2 expression data. In all 5 embryos subjected to in situ hybridization, the cells in the lower part of the fissure are negative for vsx2 (5/5). However, in the dorsal region containing the pioneer cells the pattern was not as clear. In 2/5 cases the staining was not reaching the level of the pioneer cells.  In 1/5 cases the upper temporal side was vsx2 negative and the upper nasal side positive potentially reaching the pioneer cells (e.g. Figure 5K). In 2/5 cases vsx2 expression was detected on both sides potentially reaching the level of the pioneer cells (e.g. Figure 3G). Thus, we are more cautious with the notion that the pioneer cells are vsx2 negative.

For dct as a marker for the RPE precursors we had subjected 7 embryos to the in situ hybridization. Here, we found dct expression in the “RPE precursor” cells regularly in the RPE domain. In the fissure region we found the dct expression in the lower part of the optic fissure, but not reaching the pioneer cells (6/7), 1/7 unclear if level of pioneer cells is reached.

We included this information to the main text. Please see the revised section 2.3.

  Comparison of vsx2 expression vs BMP induction again attempts to claim a difference in vsx2 expression in the fissure margins, however, the images in the figures are not convincing. Figure 5 K vs L shows optical planes that are not exactly comparible, control images display a significantly smaller lens region and lack the level of apposition of the retinal lobes observed in the BMP treated sample.

Lastly, I do not see a clear difference in the expression of vsx2 amongst the treatments at the resolution used. Is the arrow in panel K1 suggesting that the entire lobe is missing vsx2 expression (this was not observed in figure 3)? The images are of low quality and therefore difficult to determine presence or absence of expression in specific cells.  In order to be valid, all the vsx2 expression data needs to be quantified, simple qualitative observation comparison is too subjective.

We exchanged the image and now provide an optical plane which is more distal. We also replaced it with a close up of the fissure margins. An overview is provided in Figure 3 (different embryo). We are confident that the changes resulting from BMP induction are visible better now, being an increase in vsx2 in the fissure domain and an eversion of dct expression.

Figure 7: The authors need to show comparable images for their treatments. Fig 7H lacks an accompanying control image (max intensity projection of the control embryo)

We included a z-projection of the dorsal retinal vessel in the new supplemental Figure 7. Here, we manually removed all superficial endothelial cells that were also GFP positive to show the only the vessels. Please see the Figure Supplement 7 and the respective Figure legend.

Minor point.

The authors indicate that their and other work have found that fusion initiates at ~37hpf, yet their laminin data, Figure 5, suggests that up to 42hpf the basement membrane is intact. Does this indicate that pioneer cell interaction, and therefore initial fusion proceeds without basement membrane removal? The authors should talk about this discrepancy between their and published work.  

We thank the referee for this notion. The embryos were subjected to a heat shock, which results in  delay of development.  Independent of this, we find it likely that the pioneer cells are actively approaching each other even before the basal lamina is dissolved.  

We added a notion in the text in line 244 f “Taking the information from our in vivo analyses and from our analyses on processed embryos into account it seems likely that the pioneer cells are actively approaching each other with the basal lamina being still intact (Figure 2 Supplement 2).”

Reviewer 2 Report

In vivo analysis of optic fissure closure in zebrafish 

In this nicely written manuscript, Eckert and colleagues use an elegant “4D imaging” system to look at the dynamics of optic fissure closure in zebrafish and to explore the role of bone morphogenetic proteins (BMPs) in the process.  They identify so-called “pioneer cells”—the first cells in the fissure to make contact—in the proximal optic cup that initially seem to have properties distinct from retinal pigment epithelium (RPE) and neural retina (NR) precursors by marker expression, but that eventually incorporate into the NR.  Timed induction of BMP results in non-closure of the fissure (i.e., coloboma), likely due to a local process rather than a global morphogenetic one. 

The pictures and movies and stunning and the authors raise some intriguing hypotheses based on these observations. 

Comments are given with the prefix “L” for denote a line number in the draft I received. 

SCIENTIFIC 

  1.  L55.  The significance of the cloche mutant may not be known to some readers; consider briefly expanding.  Also, please comment here or elsewhere whether/how angiogenesis vs. vasculogenesis are affected in these mutants.  
  1. L59.  Similar comment for lmo2.  The specific significance of this mutant may be lost on the reader.   
  1. L127, “…drives expression initially in all retinal progenitor cells and is subsequently restricted to progenitors of the NR.”  Is the point here that it is initially in progenitors of both RPE and NR (i.e., the generic “retina”) or is this a mistake in phrasing?  (Off hand, “retinal progenitor cells” and “progenitors of the NR” might be considered synonymous by some.) 
  1. L239.  The authors’ point that these pioneer cells appear neither to be committed NR or RPE is intriguing and the switch to rx2 positivity is a nice observation.  I think one fly in the ointment is that most presumptive RPE morphologically starts non-pigmented.  Pigmentation proceeds from dorsal to ventral, ending around the time of optic fissure closure.  Is it possible that the pioneer cells are really such cells that just haven’t yet turned on their pigment genes at this time?  Please discuss. 
  1. Circa L288.  The authors should consider referencing the very recent publication from the Kwan lab (PMID:31988185) that I think complements their findings of the roles of periocular mesenchyme and basement membrane dissolution.  Consider discussing/contrasting here or elsewhere in the paper. 
  1. L330ff.  Consider adding a statement as to the expression pattern of receptors that bind to BMP4 and whether these are on optic cup/fissure cells and/or POM. 
  1. Figure 5: panels K-L.  I am having trouble seeing the local increase in vsx2 expression the authors refer to.  Is it possible to have an enlarged image of the WT embryos as is shown for the BMP4-induced embryos please to assist with this? 
  1. L404.  Please state whether this sox10 cis-regulatory element drives expression in both neural crest and mesoderm cells (the components of POM), if known. 
  1. Figure 7: I am a little confused, if lyntdTomato is supposed to ubiquitously label cells, why more cells are not outlined, especially in the control panels.  Also, can the dorsal retinal vessel be identified in one of the control panels as well? 

EDITORIAL 

  1. L44.  The paragraph beginning in this line is long and contains important information about multiple subtopics in the paper.  Consider breaking into two paragraphs: basal lamina and BMP signaling.  
  1. L128, consider changing to, “ubiquitous cell membrane labelling…” 
  1. L146, not sure if “glial” should be capitalized 
  1. L239, consider including the full name, “dopachrome tautomerase as well 

Author Response

Reviewer 2 

In vivo analysis of optic fissure closure in zebrafish 

In this nicely written manuscript, Eckert and colleagues use an elegant “4D imaging” system to look at the dynamics of optic fissure closure in zebrafish and to explore the role of bone morphogenetic proteins (BMPs) in the process.  They identify so-called “pioneer cells”—the first cells in the fissure to make contact—in the proximal optic cup that initially seem to have properties distinct from retinal pigment epithelium (RPE) and neural retina (NR) precursors by marker expression, but that eventually incorporate into the NR.  Timed induction of BMP results in non-closure of the fissure (i.e., coloboma), likely due to a local process rather than a global morphogenetic one. 

The pictures and movies and stunning and the authors raise some intriguing hypotheses based on these observations. 

 We want to thank the referee for the encouraging comment and for the constructive comments below.

Comments are given with the prefix “L” for denote a line number in the draft I received. 

SCIENTIFIC 

  1.  L55.  The significance of the cloche mutant may not be known to some readers; consider briefly expanding.  Also, please comment here or elsewhere whether/how angiogenesis vs. vasculogenesis are affected in these mutants.  

We want to thank the referee for pointing this out. We included a brief explanation of the cloche mutant fish into L55 ff and following. This now reads: “. Interestingly, the lack of the early hyaloid vessels addressed in cloche mutants in which the vasculogenesis is impaired due to failure in endothelial specification resulting from a mutation of the master regulator Npas4l ( Stainier et al., 1995, Thompson et al., 1998, Reischauer et al., 2016, Maras et al., 2019) did delay, but not prevent basal lamina dissolution [23].

The Citations have been included as numbers though. This changed the numbering of the overall references.

  1. L59.  Similar comment for lmo2.  The specific significance of this mutant may be lost on the reader.   

We also thank the referee for this suggestion and are happy to describe the lmo2 a bit further. This now reads: “On the other hand, it was described that an enlarged hyaloid vessel, found in lmo2 zebrafish mutants, was hampering fissure fusion [26], lmo2 being an endothelial factor needed for angiogenesis rather than vasculogenesis as Npas4l. This is interesting since lmo2 was found to be regulated by Npas4l (Yamada et al., 2000, Maras et al., 2019). The Citations have been included as numbers though. This changed the numbering of the overall references. Please see the change in line 61 f.

In addition we included in L 589 ff: “The latter, however, seems unlikely since a reduction of hyaloid vein size in the lmo2 mutant was sufficient to rescue the coloboma phenotype (Weiss et al., 2012).” The Citation has been included as number though.

  1. L127, “…drives expression initially in all retinal progenitor cells and is subsequently restricted to progenitors of the NR.”  Is the point here that it is initially in progenitors of both RPE and NR (i.e., the generic “retina”) or is this a mistake in phrasing?  (Off hand, “retinal progenitor cells” and “progenitors of the NR” might be considered synonymous by some.) 

Yes, indeed this is want we aimed to say. This is not a mistake in phrasing. We especially appreciate this notion. This is an important point to consider here and we now tried to make this point clearer. This now reads: In Line 179 “It was shown previously that an rx2 cis regulatory element of medaka (Oryzias latipes) drives expression initially in all retinal progenitor cells meaning progenitors for both NR and RPE and is subsequently restricted to progenitors of the NR [21].”

  1. L239.  The authors’ point that these pioneer cells appear neither to be committed NR or RPE is intriguing and the switch to rx2 positivity is a nice observation.  I think one fly in the ointment is that most presumptive RPE morphologically starts non-pigmented.  Pigmentation proceeds from dorsal to ventral, ending around the time of optic fissure closure.  Is it possible that the pioneer cells are really such cells that just haven’t yet turned on their pigment genes at this time?  Please discuss. 

We want to thank the referee for this comment. We are aware of the delay in pigmentation and especially the fact that the onset of pigmentation is dorsal with the pigmentation then spreading to ventral domains concerned us. We thought it unlikely to find a marker for the prospective RPE in the optic fissure domain at this stage of development. Fortunately, we identified dct as a good marker for the prospective RPE cells, preceding pigmentation and being expressed even within the margins of the optic fissure. Overall our data suggest that the pioneer cells are negative for dct, down-regulate rx2 driven GFP and are potentially positive for vsx2. Importantly, after contact formation at least a part of these cells reactivate their rx2 driven GFP and integrate into the NR. In addition, the pioneer cells were found to be rather flat. A flat morphology of contact forming cells was earlier described by Hero, 1990. From that, it was concluded that these cells are RPE precursors. Interestingly, our observation is at odds with but also extending these previous but lasting assumptions that RPE precursors are establishing the contact to initiating the fusion. We propose and show data supporting this that the contact forming cells are a distinct population of cells.

  1. Circa L288.  The authors should consider referencing the very recent publication from the Kwan lab (PMID:31988185) that I think complements their findings of the roles of periocular mesenchyme and basement membrane dissolution.  Consider discussing/contrasting here or elsewhere in the paper. 

We thank the referee for pointing towards this study (Bryan et al., 2020) from the Kwan lab. This is in fact a very nice analysis, however, rather focusing on the role of the POM during the morphogenesis of the optic fissure than on the role of the POM during its fusion. We thus, in order to not confuse the reader, chose not to include it.

  1. L330ff.  Consider adding a statement as to the expression pattern of receptors that bind to BMP4 and whether these are on optic cup/fissure cells and/or POM. 

Based on own unpublished data, limited so far to in situ hybridizations of one BMPRI we assume that the receptor is expressed in the neuroectodermal tissue rather than in the POM. We cannot exclude an expression within the POM though. Data presented by Alexander et al 2014 showed expression within the neuroectodermal tissue (supplemental material). An expression in the POM, from our perspective, can also not be ruled out there.

  1. Figure 5: panels K-L.  I am having trouble seeing the local increase in vsx2 expression the authors refer to.  Is it possible to have an enlarged image of the WT embryos as is shown for the BMP4-induced embryos please to assist with this? 

We thank the referee for pointing this out. We now provide a better visualization. Please see the revised Figure 5.

  1. L404.  Please state whether this sox10 cis-regulatory element drives expression in both neural crest and mesoderm cells (the components of POM), if known. 

To our knowledge, the sox10 cis regulatory element is driving expression in the neural crest derived POM. We included the following into Line 710 f: “ Sox10 is an established marker for the neural crest derived cells  rather than for other mesenchymal cells of the POM (Dutton et al., 2001, Langenberg., et al., 2008, Bryan et al., 2020.)” The Citations have been included as numbers though. This changed the numbering of the overall references.

  1. Figure 7: I am a little confused, if lyntdTomato is supposed to ubiquitously label cells, why more cells are not outlined, especially in the control panels.  Also, can the dorsal retinal vessel be identified in one of the control panels as well? 

Here, for imaging we did not introduce PTU to the zebrafish medium. Therefore pigmentation started especially in the dorsal part of the eye, leading to a weaker signal of the lyntdTomato. Furthermore, an irregular pattern could be explained due to an RNA injection for lyntdTomato.

We are able to identify the dorsal retinal vessel in our imaging data and we labeled it (Figure 7D). We also included a z-projection of the dorsal retinal vessel in the Figure supplements. Here we manually removed all superficial endothelial cells that were also GFP positive to show the only the vessels intended. Please the new Figure 7 Supplement and the respective legend.

EDITORIAL 

  1. L44.  The paragraph beginning in this line is long and contains important information about multiple subtopics in the paper.  Consider breaking into two paragraphs: basal lamina and BMP signaling.  

We thank the referee for this suggestion. We are happy to have changed this. Please see the introduced break in Line 70.

  1. L128, consider changing to, “ubiquitous cell membrane labelling…” 

This was changed to: ”We used the transgenic line tg(Ola.rx2:egfp-caax) [21] and a ubiquitous cell membrane labeling by…” Please see Line 181 in the revised version.

  1. L146, not sure if “glial” should be capitalized 

We changed it and lowercased it. Please see Lines 200 and 202 in the revised version.

  1. L239, consider including the full name, “dopachrome tautomerase” as well 

We included the full name, please the revised Line 348.

Round 2

Reviewer 1 Report

After several rounds of revision I feel the authors have now satisfied my concerns. I appreciate their efforts and commend their dedication to the work.

This manuscript is a resubmission of an earlier submission. The following is a list of the peer review reports and author responses from that submission.

Round 1

Reviewer 1 Report

In this paper, Eckert et al., describe choroid fissure closure  in the zebrafish and identify a role for BMP signaling in preventing basement membrane breakdown and in regulating hyaloid vasculature.

Overall Comments:

The paper is conceptually straightforward, and its main strength is in 4-D imaging. It is largely descriptive and some of its main findings have been reported elsewhere (e.g., pioneering cells, a role for BMP signaling in preventing basement membrane breakdown, and in producing persistent hyaloid vasculature ). Choppy writing and editing and incorrect citations make the paper difficult to read.In many cases, the methods are poorly described, and the finding are over-interpreted.  The rationale for some quantitative analyses is not provided and statistics analyses do not appear to have been conducted. 

Detailed Comments:

Fig. 1.

More markers are needed to substantiate the claim that Rx2 exclusively labels neural retina. For example, in Fig. 1D (nasal, the RPE appears to be RX2 (GFP)+. Furthermore, the reacquisition of Rx2 expression by the pioneering cells does not necessarily indicate a RPEàNR conversion.  It could indicate a switch in the state of the NR.

The claim that continuity between the NR and the RPE is established by pioneering cell mediated CF fusion and contribution to both tissues is not supported by the images in Figure 1. Again, retinal and RPE markers and high resolution image and fate-mapping are needed to substantiate this claim. 

The cartoon (1K-N) depicts events not described in the results (e.g., cell division of pioneering cells, Fig. 1N).

Fig. 2. 

How are the authors determining the identity of the hyaloid artery and vein? Fig. 2 does not provide evidence that the hyaloid vain is moving distally due to CF fusion. The images in 2E, F are fuzzy and hard to evaluate. In the schematic, proximal is to the left.

Fig. 3.

The schematic in A, does not adequately describe the quantitation paradigm.This is not described in the text either until the methods section.  The significance of this quantitation is not clear.  The fact the CF fuses dorsally first has been described in previous papers, so this does not appear to add any new data.  This figure should be eliminated. Lines 209-212.It’s not clear what the authors mean by this statement.  Why does the dorsal contact of pioneering cells mean that the NR, RPE, pioneering cells are brought into position late in optic fissure morphogenesis? “Late” should be defined. A number of citations are incorrect in the paragraph beginning with line 206.E.g., Ref 36, line 217.  No statistical data are provided for any quantitation in the paper.

Fig. 4.

Why are the authors ruling out growth defects as an underlying cause of the coloboma as shown before by several authors (e.g., Morcillo et al., 2006)? The persistence of the basal lamina has been shown in BMP loss of function experiments in chicks.How do the authors reconcile their findings with the opposite results in the chick? 

Fig. 5.

What times do the schematics in K, right panel represent?The controls do not look like I or J, although the BMP induced schematic looks like D, E. Also, the images in I, J are hard to interpret due to poor resolution.

Other Comments: 

The Introduction does not provide any background for the role of BMP signaling. In particular, the authors findings have overlap with recent studies examining BMP function in producing persistent hyaloid vasculature. The authors findings are not placed in the context of these findings.  A time of CF closure and POM/hyaloid vasculature needs to be provided.

Author Response

Dear Reviewer,

at first, we want to thank for the constructive comments on our work, which we highly appreciated. Due to the limited time for revising our work, we mainly focused on the editorial work on our manuscript and the data presentation.

We realized, that the description and presentation of our data needed further improvement. We enclosed the new version of the manuscript, which we carefully revised. The changes are highlighted via the “track changes” modus.

Please find enclosed our answers to your comments, point by point.

In this paper, Eckert et al., describe choroid fissure closure  in the zebrafish and identify a role for BMP signaling in preventing basement membrane breakdown and in regulating hyaloid vasculature.

Overall Comments:

The paper is conceptually straightforward, and its main strength is in 4-D imaging. It is largely descriptive and some of its main findings have been reported elsewhere (e.g., pioneering cells, a role for BMP signaling in preventing basement membrane breakdown, and in producing persistent hyaloid vasculature ). Choppy writing and editing and incorrect citations make the paper difficult to read.In many cases, the methods are poorly described, and the finding are over-interpreted.  The rationale for some quantitative analyses is not provided and statistics analyses do not appear to have been conducted. 

We want to point out, that especially the aspect of the pioneer cells, which was mentioned in a previous publication (Gestri et al 2018), was citing our previous version of the manuscript, which is available on bioRxiv. It is well possible that the citation is missing in the reference list, however, in the text our data is clearly referred to.

Detailed Comments:

Fig. 1.

More markers are needed to substantiate the claim that Rx2 exclusively labels neural retina. For example, in Fig. 1D (nasal, the RPE appears to be RX2 (GFP)+. Furthermore, the reacquisition of Rx2 expression by the pioneering cells does not necessarily indicate a RPEàNR conversion.  It could indicate a switch in the state of the NR.

We thank the referee for this comment. It is important to consider that the regions of the fissure margins consist of undifferentiated progenitors. It is also important to note that the prospective NR and RPE are situated side by side on the same basal lamina. Before the optic fissure is fused, it is unclear which cells will be integrated into the future NR and which cells will be integrated into the future RPE. However, we noticed that next to the morphological changes the prospective RPE cells undergo to obtain a flat morphology, the expression of the rx2 driven GFP was reduced to absent in these cells. 

We then carefully focused on the fusing optic fissure region with 3D in vivo imaging over time. We found that the cells, which establish the cellular contact in between the margins showed reduced GFP signal up to an almost total loss of GFP signal. This was actually the rational to include another fluorescent label, besides the rx2 driven GFP, to visualize the process of optic fissure fusion.

We agree that the reactivation could indicate a switch in the state of the NR. Nevertheless, this population of cells behaved differently than the other NR progenitors.

The claim that continuity between the NR and the RPE is established by pioneering cell mediated CF fusion and contribution to both tissues is not supported by the images in Figure 1. Again, retinal and RPE markers and high resolution image and fate-mapping are needed to substantiate this claim. 

We agree with the referee, that fate mapping would be the best way to address this aspect. This, however, would need a not yet available marker for the early RPE precursors. The

RPE differentiation starts in the dorsal domain of the optic cup and the domain of the fused optic fissure is only differentiating much later. To our knowledge there is currently no specific marker available properly identifying the early RPE progenitors. We, however, performed in vivo recordings of 3D volumes over time and used these for data analysis. The data presented in Figure 1 displays one optical section over time only, to show the partial reactivation of rx2 driven GFP and integration into the future NR and the cell which we could track into the future RPE domain.

The cartoon (1K-N) depicts events not described in the results (e.g., cell division of pioneering cells, Fig. 1N).

We thank the referee for this comment. With the scheme presented in Figure 1 (K-N) we aimed to point out that the pioneer cells could contribute to the future NR and RPE in different ways. They could do so as a population of cells, meaning that single cells are integrated into the NR and others into the RPE. However, another option could be a cell division, with the progeny being integrated in to both NR and RPE. We discussed this in the section 2.1 of the manuscript.

Fig. 2. 

How are the authors determining the identity of the hyaloid artery and vein? Fig. 2 does not provide evidence that the hyaloid vain is moving distally due to CF fusion. The images in 2E, F are fuzzy and hard to evaluate. In the schematic, proximal is to the left.

Figure 2D shows the hyaloid vein entering the optic cup through the optic fissure, while 2E shows the optic fissure margins forming contact over the vein. Figure 2F shows the fused “former optic fissure” in the same optical plane as D and E. Supplemental Movie File 4 shows the dynamics of this process.

In order to be able to fuse distal of the original position of the vein, the vein has to either move distally or be split in half. Our imaging data indicates that the vessel remains continuous and is not split throughout the fusion process. Furthermore, we recorded in 3D over time. Our data show, that the site of fusion is progressing distally and with it the hyaloid vein is moved distally. In order to improve the data presentation, especially for the latter aspect, we included a different perspective, derived from our original data. Please see the revised Figure 2.

Fig. 3.

The schematic in A, does not adequately describe the quantitation paradigm.This is not described in the text either until the methods section.  The significance of this quantitation is not clear.  The fact the CF fuses dorsally first has been described in previous papers, so this does not appear to add any new data.  This figure should be eliminated. Lines 209-212.It’s not clear what the authors mean by this statement.  Why does the dorsal contact of pioneering cells mean that the NR, RPE, pioneering cells are brought into position late in optic fissure morphogenesis? “Late” should be defined. A number of citations are incorrect in the paragraph beginning with line 206.E.g., Ref 36, line 217.  No statistical data are provided for any quantitation in the paper.

The point of the cellular contact establishment between the fissure margins was estimated, based on 3D in vivo imaging recordings over time. The point of the cellular contact is further defining the location of the pioneer cells. We included subtle changes in the scheme presented in “A” and furthermore included the description of measurement to the main text to clarify this point. In a previous analysis, we focused on optic fissure morphogenesis (Eckert et al., 2019). We showed that “bilateral flows” of tissue from the optic stalk and the lens-averted domain of the optic cup are important for the establishment of the margins. The cells which are integrated into the margins at the end of the “flow” and at the position in the upper third of the margin, will likely become the pioneer cells.

Thus, it is important to note that eventually the onset of fusion is in the upper domain of the fissure, as well as in the proximal domain, close to the future optic nerve head. No statistical analysis was performed since we measured only in the wildtype condition.

Fig. 4.

Why are the authors ruling out growth defects as an underlying cause of the coloboma as shown before by several authors (e.g., Morcillo et al., 2006)? The persistence of the basal lamina has been shown in BMP loss of function experiments in chicks.How do the authors reconcile their findings with the opposite results in the chick? 

We thank the referee for this comment. Ruling out morphogenetic defects is important for our analysis and this aspect is especially important, since BMP plays important roles during eye morphogenesis as well. We therefore established an inducible transgenic line in which we were able to express BMP4 upon heat-shock (Knickmeyer et al., 2018). We showed that this line can be used to alter optic cup and optic fissure morphogenesis (Knickmeyer et al., 2018, Eckert et al., 2019), when “induced” early. However, a heat-shock mediated induction of BMP4, if performed later, was sufficient to inhibit the fusion of the margins.

Further, the referee was probably pointing towards the paper by Adler and Belecky-Adams (2002). There the induced expression of noggin as well as a dominant-negative BMP receptor also resulted in coloboma formation and a persisting basal lamina. However, based on the data presented in this analysis, it is likely that a morphogenetic defect was achieved by the induced expression of noggin as well as the dominant negative-receptor. It is thus possible that this could be the reason for a persisting basal lamina in their analysis.

The referee is right that it is difficult to explain that BMP induction (Heermann et al., 2015, Knickmeyer et al., 2018, Eckert et al., 2019) and BMP signaling inhibition (Adler and Belecky-Adams, 2002) show potentially similar results during optic cup morphogenesis. It could thus be interesting to further investigate the differences between both, however, this is not the focus of our current manuscript.

Fig. 5.

What times do the schematics in K, right panel represent?The controls do not look like I or J, although the BMP induced schematic looks like D, E. Also, the images in I, J are hard to interpret due to poor resolution.

The schematics in K right panel represent H-J of the controls. In H the first contact of the pioneer cells over the HV is depicted. From there onwards, the NR size increases and the HV is moved ventrally (I-J)  

Other Comments: 

The Introduction does not provide any background for the role of BMP signaling. In particular, the authors findings have overlap with recent studies examining BMP function in producing persistent hyaloid vasculature. The authors findings are not placed in the context of these findings. 

We thank the referee for this comment. We included more background information for BMP signaling and eye development into the introduction.

Then the referee is likely pointing towards the publication of Hocking et al., 2018. We thank the referee for pointing this out. We were happy to include it into our manuscript.

A time of CF closure and POM/hyaloid vasculature needs to be provided.

We now refer also to the timing of fissure fusion. Please see the revised section 2.3.

Reviewer 2 Report

In summary, Eckert et al provide data of the process of optic fissure closure in zebrafish derived mostly from in vivo imaging. Their data shows that contact initiation involved a population of non-rx2 expressing cells in the fissure margin that fuse to initiate optic fissure closure, and that this process physically separates the hyaloid artery and veins. The continued fusion of the optic fissure then drives the distal displacement of the hyaloid vein throughout the fusion process. The pioneer cells appear to regain rx2:gfp expression after fusion has been initiated and may contribute to the fused epithelia. The authors also provide data to establish the points of initiation along the DV and PD axes, and also show the integrity of the basement membrane is disrupted at these points, coincident with fusion. The authors then show that genetic overexpression of bmp4 during optic fissure closure causes disruption to the development of the hyaloid vasculature. The work regarding fusion initiation complements and largely confirms observations made in a recent report by Gestri and colleagues (Gestri et al., 2018; doi: 10.3389/fncel.2018.00042), however the hyaloid vasculature observations appear to be novel. In general this is a minor addition to the field of optic fissure closure biology, that could however be greatly improved by further work on both content and presentation.

Major points:

Much of the text throughout the manuscript is written poorly, both grammatically and scientifically, and improving this would provide more clarity to judge the scientific merits of the work. Indeed, ultimately it is difficult to judge the conclusions of the work.

The abstract is confusing and requires more accurate statements of what the study actually found. For example, line 18 states the “pioneer cells leave the epithelia through the basolateral aspect…” however the polarity of these cells is not clearly described in the data presented, and the data shows these cells do not “leave” the epithelia, rather they extend as a group towards the opposing margin. There is also insufficient data showing these cells to be “plastic”. The abstract does not mention the development of the hyaloid vasculature in normal contexts during optic fissure closure, for which there are several main figures describing this process.

The introduction is unfocused and confusing, failing to set up the study adequately. It would be good to clarify which mechanisms are currently poorly understood, and which are specifically being addressed in this study. The authors could consider focusing more on the emergence of pioneer cells and epithelial cell movements in other fusion contexts and species, for example (see point 2, below). Line 65 appears to suggest the paper will investigate proliferation and cell death and separate these from fusion processes, however these mechanisms are not covered in the study. The authors state (lines 70-71) that they “address importantly, how the contact between in between margins is established and how an epithelial disassembly of the margins is followed” – the study indeed shows where the contact is initiated, and goes some way to show which cells initiate this contact, however there is no direct evidence showing the process of epithelial disassembly. The authors should address this to substantiate this claim.

The identification of pioneer cells is not novel and was recently shown by Hardy et al (2019) and the same cells, although not referred to as pioneer cells, were found by Gestri et al (2018). Therefore, this study should take the chance to define these cells in much more detail than is currently shown, as this would represent a major step forward in the field.

Relying on the rx2:egfp data as the sole line of evidence is insufficient to determine pioneer cell characteristics, their origin, or fate in the fused margin. Immunofluorescence for additional RPE- and NR-specific markers would more accurately define these cells, as would markers for epithelial cells and for cell-polarity (as shown by Gestri et al) to show their orientation changes during contact and fusion. Indeed, higher magnification images would be useful to show the nature of the first contact. Can the authors confirm this is specifically made between opposing RPE cells, or are other cells involved?

The breakdown of the BM has already been observed in chick, mouse and zebrafish in various recent and historical studies. Can the authors elaborate on the importance of showing this same process in their work?

Minor points:

The tense of the writing changes multiple times throughout the manuscript. The numbers of observations is not consistently mentioned for each experiment and in some instances statistical analyses are required. There is no reference to many of the Supplemental movies in the main text. These should be described and adequately referred to. In Figure 1, the tissue architecture at the point of fusion is different to that shown in Gestri et al. Can the authors provide an explanation for this observation? The temporal fissure margin appears to contribute the majority of the initiating cells but is not mentioned. How consistently was this observed? To confirm the data suggesting optic fissure fusion acts to displace the hyaloid vein (Figure 2), sagittal sections are also required. Markers of POM, hyaloid vessels and neural retina precursors would be useful to determine if the contact between the pioneer cells and HV also requires POM. Data in figure 3A-C could be better supported by including the size of the hyaloid vein, as this appears to be a critical factor in defining the fusion initiation point and would certainly influence its position. In addition, it would be a useful addition to show how the fusion progresses in the PD axes at timepoints subsequent to the initial contact. Figure 3. How was each fusion point unambiguously defined in these analyses? The conclusions from Figure 3D could be improved by showing the position of the initial contact point relative to the axial length of the eye, rather than the other fixed landmarks, and including measurements of whole proximal-distal length or total volume of the eye cup. Line 55. The recent study by Hardy et al [ref 25] did not show netrin to breakdown the basement membrane during chick optic fissure, they showed that netrin expression coincided with BM breakdown. Please amend this sentence to ensure accuracy of citations. What does “experimentally induced BMP mean? This needs to be clearly explained and references cited. Line 247. “It can be appreciated that the experimental induction..” appreciated by who? Please make this statement accurately reflect the data, e.g. use “We observed..” Fig 4. The authors state there is no difference to the tissue proceeding fusion in the bmp4 induced fissures, however it appears as if the nasal and temporal margins have indeed changed their morphologies, with the temporal margin more rounded and the nasal fissure more pointed. Is this consistent in all samples analysed? This is different to the control fissures and to figs 1 and 2. How many embryos showed defective fusion in the cohort tested? Is fusion delayed and does it occur subsequently as bmp is reduced? The intensity of the Rx2:gfp is lower in the tg (hsp:bmp4) -Figure 4A-D. Is there a reason for this? Line 385. What is the relevance of the control embryos “without a beating heart”? The laminin fluorescence intensity of the entire basement membrane is reduced in Figure 4P and is variable throughout the panels. High magnification images of the BM could support the claim that it is not broken down when bmp is induced. In movie 5 there appear to be filopodial extensions observed between he opposing margins, however the basement membrane data presented in. However, Figure 4M-P suggests there is no breakdown of BM. Can the authors clarify what may be occurring, for example is this evidence of invadopodia formation? This would be very interesting. Although the bmp embryos are heat shocked, there is no evidence to show whether this is active, e.g. through pSMAD signalling. It would also be useful to see in which optic fissure cells the pSMAD is active in response to bmp. The exposure is saturated in the beginning of Movie 5, and the timings are not shown. These should be addressed to enable clearer interpretation of this data. Figure 5. The data suggests that bmp4 prevents separation of the HA/HV, however it is not stated how many embryos this observation was seen in or the total embryos/eyes used for analysis. The data could also be improved through quantitation for the fluorescent signal against background and could prevent artefacts from mis-aligned optical sections and improve the comparison between bmp and control groups as the qualitative difference appears subtle in the figure presented. The term “abnormal hyaloid vein” needs more specific description, e.g. is it “abnormally located”? Can the authors confirm the fli1a:gfp signal observed in Fig6B are hyaloid capillaries and not presumptive HA or HV? There is green signal visible in the centre region of Fig6B - are these proximally located HA/HV or are these distal tissues? Line 328. “..site of fusion is eventually displacing the hyaloid vein towards the periphery”. This is a clumsy sentence, while there is also no direct evidence showing this occurring in the study. Can the authors show a sagittal view of this happening to help confirm this? Could the heat shock be delayed to arrest the progression of fusion after initiation. This would help show that fusion progression is essential for driving the HV distally, as the HV should be in an intermediate position adjacent to the halted region of fusion?

Author Response

Dear Reviewer,

at first, we want to thank for the constructive comments on our work, which we highly appreciated. Due to the limited time for revising our work, we mainly focused on the editorial work on our manuscript and the data presentation.

We realized, that the description and presentation of our data needed further improvement. We enclosed the new version of the manuscript, which we carefully revised. The changes are highlighted via the “track changes” modus.

Please find enclosed our answers to your comments, point by point.

In summary, Eckert et al provide data of the process of optic fissure closure in zebrafish derived mostly from in vivo imaging. Their data shows that contact initiation involved a population of non-rx2 expressing cells in the fissure margin that fuse to initiate optic fissure closure, and that this process physically separates the hyaloid artery and veins. The continued fusion of the optic fissure then drives the distal displacement of the hyaloid vein throughout the fusion process. The pioneer cells appear to regain rx2:gfp expression after fusion has been initiated and may contribute to the fused epithelia. The authors also provide data to establish the points of initiation along the DV and PD axes, and also show the integrity of the basement membrane is disrupted at these points, coincident with fusion. The authors then show that genetic overexpression of bmp4 during optic fissure closure causes disruption to the development of the hyaloid vasculature. The work regarding fusion initiation complements and largely confirms observations made in a recent report by Gestri and colleagues (Gestri et al., 2018; doi: 10.3389/fncel.2018.00042), however the hyaloid vasculature observations appear to be novel. In general this is a minor addition to the field of optic fissure closure biology, that could however be greatly improved by further work on both content and presentation.

We want to point out, that especially the aspect of the pioneer cells, which was mentioned in a previous publication (Gestri et al 2018), was citing our previous version of the manuscript, which was and is available on bioRxiv. It is well possible that the citation is missing in the reference list, however in the paper it was clearly referred to our data.

Major points:

Much of the text throughout the manuscript is written poorly, both grammatically and scientifically, and improving this would provide more clarity to judge the scientific merits of the work. Indeed, ultimately it is difficult to judge the conclusions of the work.

We carefully revisited the manuscript to make it more accurate. We hope that this provided more clarity to follow our conclusions and to judge the merit.

The abstract is confusing and requires more accurate statements of what the study actually found. For example, line 18 states the “pioneer cells leave the epithelia through the basolateral aspect…” however the polarity of these cells is not clearly described in the data presented, and the data shows these cells do not “leave” the epithelia, rather they extend as a group towards the opposing margin. There is also insufficient data showing these cells to be “plastic”. The abstract does not mention the development of the hyaloid vasculature in normal contexts during optic fissure closure, for which there are several main figures describing this process.

We carefully changed the abstract. Furthermore, we deleted the word plastic from the title as well as from the abstract. We thank the referee for pointing out the unclear wording “pioneer cells leave the epithelia through the basolateral aspect…”, which we were happy to change.

The introduction is unfocused and confusing, failing to set up the study adequately. It would be good to clarify which mechanisms are currently poorly understood, and which are specifically being addressed in this study. The authors could consider focusing more on the emergence of pioneer cells and epithelial cell movements in other fusion contexts and species, for example (see point 2, below). Line 65 appears to suggest the paper will investigate proliferation and cell death and separate these from fusion processes, however these mechanisms are not covered in the study. The authors state (lines 70-71) that they “address importantly, how the contact between in between margins is established and how an epithelial disassembly of the margins is followed” – the study indeed shows where the contact is initiated, and goes some way to show which cells initiate this contact, however there is no direct evidence showing the process of epithelial disassembly. The authors should address this to substantiate this claim.

We also carefully revised the introduction. We included input also from the other referees. We hope that the study is set up adequately now by the introduction. We, however, maintain the statement of an “epithelial disassembly”, since the cellular contact establishment as well as the dissolution of the basal lamina is essentially an epithelial disassembly. The referee is right, that we did not include markers for the cell polarity, as performed by Gestri et al., also pointed out below. Nevertheless, it is clear that the pioneer cells extend through the basolateral side of the epithelium, thereby initiating the contact between the margins and as a consequence disassemble the original margins.

The identification of pioneer cells is not novel and was recently shown by Hardy et al (2019) and the same cells, although not referred to as pioneer cells, were found by Gestri et al (2018). Therefore, this study should take the chance to define these cells in much more detail than is currently shown, as this would represent a major step forward in the field.

We want to point out, that especially the aspect of the pioneer cells, which was mentioned in a previous publication (Gestri et al 2018), was citing our previous version of the manuscript, which is available on bioRxiv. It is well possible that the citation is missing in the reference list, however in the text it is clearly referred to our data. The recent publication by Hardy et al., to our knowledge did not specifically refer to pioneer cells.  To prevent further problems with the citation of our primary identification of the pioneer cells, we included a citation of our bioRxiv manuscript.

Relying on the rx2:egfp data as the sole line of evidence is insufficient to determine pioneer cell characteristics, their origin, or fate in the fused margin. Immunofluorescence for additional RPE- and NR-specific markers would more accurately define these cells, as would markers for epithelial cells and for cell-polarity (as shown by Gestri et al) to show their orientation changes during contact and fusion. Indeed, higher magnification images would be useful to show the nature of the first contact. Can the authors confirm this is specifically made between opposing RPE cells, or are other cells involved?

We agree with the referee, that more markers for the identification of the pioneer cells would be helpful and that in order to investigate the origin and fate of these pioneer cells, would require a proper fate mapping. The latter is not in the scope of our current analysis and unfortunately, no specific markers for the pioneer cells are available. We do hope, however, that the thorough transcriptional analysis e.g. performed by Hardy et al., 2019 will help in that respect for future analyses.

We also agree that additional markers for precursors, most importantly for the RPE precursors, would be beneficial. RPE differentiation starts in the dorsal domain of the optic cup and the domain of the fused optic fissure is only differentiating much later. To our knowledge there is currently no marker available, properly and specifically identifying the early RPE progenitors located in the margins before fusion or directly after.

We also agree with the statement: “higher magnification images would be useful to show the nature of the first contact”. Even though we use good optics, we are limited with respect to spatial resolution. Super resolution imaging for our in vivo analysis is not feasible. To improve image quality, we tried deconvolution (Huygens) on a set of data. This, however, failed to provide the expected benetfit.

We want to point out that we do not state that the contact was initiated by the RPE cells or that the pioneer cells are a subpopulation of theses. The referee is raising an interesting point. It is described that also POM cells play a major role during optic fissure fusion. They are likely also important for basal lamina dissolution (James et al., 2016). With respect to the contact initiation, we cannot total rule out the presence of POM processes in the vicinity of the contact formation.

The breakdown of the BM has already been observed in chick, mouse and zebrafish in various recent and historical studies. Can the authors elaborate on the importance of showing this same process in their work?

The dissolution of the basal lamina is an important step during optic fissure fusion. The rational was to investigate whether the basal lamina dissolution and the cellular contact formation between the margins occur in the same region or not. Additionally, the basal lamina was persisting after BMP has been induced. This is another reason for including the basal lamina in our manuscript.

Minor points:

The tense of the writing changes multiple times throughout the manuscript.

We carefully revisited the manuscript and unified the tense.

The numbers of observations is not consistently mentioned for each experiment and in some instances statistical analyses are required.

We apologize for not having provided this information initially. The number of embryos supporting our data is now included in the manuscript. We did not perform statistical analysis, since we performed the measurements in controls only.

There is no reference to many of the Supplemental movies in the main text. These should be described and adequately referred to.

We apologize for our mistake. The supplemental movie files are an important part of our analyses and are now adequately referred to in the manuscript.

In Figure 1, the tissue architecture at the point of fusion is different to that shown in Gestri et al. Can the authors provide an explanation for this observation? The temporal fissure margin appears to contribute the majority of the initiating cells but is not mentioned. How consistently was this observed?

The referee is likely pointing towards the Figure 1 of Gestri et al., 2018. The differences in the appearance of the fissure margins are likely due to the differences of tissue treatment and can be seen even within that very figure. In B-E e.g. fixed tissue is depicted, whereas in e.g. in I living tissue is shown.

Considering our complete data sets we cannot confirm that the temporal margin is contributing the majority of cells.

To confirm the data suggesting optic fissure fusion acts to displace the hyaloid vein (Figure 2), sagittal sections are also required. Markers of POM, hyaloid vessels and neural retina precursors would be useful to determine if the contact between the pioneer cells and HV also requires POM.

We performed in vivo imaging to obtain sagittal optical sections of 3D volumes over time, presented in Figure 2 A-F. The scheme presented in Figure 2 G is presented in a transverse section to make it more intuitive to understand. We apologize, if this was causing trouble. The orientation of the images is given in the respective legend. In order to improve the data presentation, especially for the latter aspect, we included a different perspective, derived from our original data. Please see the revised Figure 2.

Regarding the potential involvement of the POM during contact initiation, the referee is raising an interesting point, already mentioned above. Currently, we cannot totally rule out the presence of POM processes in the vicinity of the contact formation. This is an interesting aspect for future analysis, beyond the scope of this revision. A discrimination between POM and hyaloid vessels will be problematic, since the POM contributes substantially to the hyaloid vasculature, please compare to Figure 5.

Data in figure 3A-C could be better supported by including the size of the hyaloid vein, as this appears to be a critical factor in defining the fusion initiation point and would certainly influence its position. In addition, it would be a useful addition to show how the fusion progresses in the PD axes at time-points subsequent to the initial contact.

We measured the size, diameter, of the hyaloid vein and included the data into Figure 3. Please see the revised Figure 3.

Figure 3. How was each fusion point unambiguously defined in these analyses? The conclusions from Figure 3D could be improved by showing the position of the initial contact point relative to the axial length of the eye, rather than the other fixed landmarks, and including measurements of whole proximal-distal length or total volume of the eye cup.

The data presented in Figure 3 is derived from measurements performed in datasets recorded via 3D in vivo imaging over time. In Figure 3 A, the fusion point was defined by the cellular contact formation of the pioneer cells. We revised the figure and now display measurements performed in distal compartments (distal from “LP”, Figure 3) only. Please see the revised Figure 3.

In Figure 3 B, the contact point was defined as the point in which the very first cellular contact in between pioneer cells from both margins was established.

We thank the referee for also suggesting an alternative way of data presentation. Relating the contact point to the total length of the eye or maybe to the length of the fissure would indeed be another option, which we discussed. Measuring the total length of the eye and relating the contact point to this is, however, likely more prone to errors. The margins are not fully aligned in distal domains causing problems during measurements and even a slight tilt during embedding for imaging could cause even more severe errors. The proximal domain of the fissure is rather fix and thus suited better.  

Line 55. The recent study by Hardy et al [ref 25] did not show netrin to breakdown the basement membrane during chick optic fissure, they showed that netrin expression coincided with BM breakdown. Please amend this sentence to ensure accuracy of citations.

We are sorry for the misunderstanding. We stated: “Recent data suggest that Netrin, a factor localized to the basal lamina, is also involved in its degradation [25]…” in our manuscript. We now rephrased this to be more accurate.

What does “experimentally induced BMP mean? This needs to be clearly explained and references cited.

The “experimental induction” we were referring to was performed in Knickmeyer et al., 2018, which we cited in this section.  We are sorry, that we missed to include the citation again after that sentence. The description of the induction procedure was included in the consecutive sentences.

Line 247. “It can be appreciated that the experimental induction..” appreciated by who? Please make this statement accurately reflect the data, e.g. use “We observed..”

We rephrased this.

Fig 4. The authors state there is no difference to the tissue proceeding fusion in the bmp4 induced fissures, however it appears as if the nasal and temporal margins have indeed changed their morphologies, with the temporal margin more rounded and the nasal fissure more pointed. Is this consistent in all samples analysed? This is different to the control fissures and to figs 1 and 2. How many embryos showed defective fusion in the cohort tested? Is fusion delayed and does it occur subsequently as bmp is reduced?

We thank the referee for this comment. The morphology of the optic fissure margins is comparable in the control and the bmp4 induced embryos at the “onset of fusion”. Please compare Figure 4 A and Figure 4 I. The referee, however, is pointing towards an interesting aspect regarding the behavior of the margins in the unfused condition, resulting from bmp4 induction, which can be seen e.g. in Figure 4 D.

The coloboma resulting from late BMP4 overexpression (heat shock post 24hpf) was described by us in Knickmeyer et al. 2018. We found that 100% of transgenic embryos subjected to heat shock had persisting optic fissures at 3-3.5 dpf. Even at 5 dpf, we never observed delayed fusion (unpublished data). In the current study, we addressed 13 heat-shock induced and 4 control embryos. At the onset of imaging (30 hpf) we found in the heat shock induced embryos 12 cases in which the margins had a WT phenotype. In one embryo the temporal fissure margin remained GFP positive, whereas the nasal margin was comparable to a wildtype. It is likely that this embryo was slightly younger.

In 12 embryos, we can clearly state that the fissure margins did not fuse. In one of the 13 embryos it remained unclear. In 2 cases the margins did not touch. In 11 cases they touch. The aberrant morphology of the margin, the reviewer observed is only occurring well after the fusion onset would have occurred. Only 4 of the 13 embryos were recorded that long. In the other 3 cases we were able to observe touching margins but not the pointed nasal margin. The control embryos were developing as expected.

The intensity of the Rx2:gfp is lower in the tg (hsp:bmp4) Figure 4A-D. Is there a reason for this?

The recordings were performed using a double fluorescent label, rx2 driven GFPcaax as well as lyntdTomato, which was introduced by mRNA injections into the zygote. The relative intensity of the Tomato signal is most likely the major factor. High levels of Tomato can pretend a low signal of GFP. Another reason could be inter-individual expression intensity variations in the tg(rx2:GFPcaax) line.

Line 385. What is the relevance of the control embryos “without a beating heart”?

We assume that the referee wanted to point to the “bleeding heart”. We apologize for using this term in the manuscript. For transgenic line generation, a transgenesis marker was used, cmlc2:GFP/myl7:EGFP. This marker resulting in a “green heart” shows the presence of the “hsp70:BMP4” construct. The tg(hsp70:bmp4) fish were outcrossed to either WT, tg(rx2:gfpcaax), or tg(sox10:lyntdTomato) and the whole clutch was heat shocked at 26hpf. Only the ones with a green heart had induction of bmp4 the ones without a green heart were used as control embryos. Importantly these control embryos also received the heat-shock.

The laminin fluorescence intensity of the entire basement membrane is reduced in Figure 4P and is variable throughout the panels. High magnification images of the BM could support the claim that it is not broken down when bmp is induced.

The anti-laminin immunohistochemistry was performed in a whole mount procedure using embryos at different ages. The staining intensity in deeper areas of the tissue (e.g. older embryos) is reduced, as referred to by the referee. We thank the referee for point this out. We thus increased the signal of the images of P and T.

We would like to keep the display of M-T. These images show a comparable section to the upper part of the figure. We are confident that the details of the persisting basal lamina in M-P are yet visible.

In movie 5 there appear to be filopodial extensions observed between he opposing margins, however the basement membrane data presented in. However, Figure 4M-P suggests there is no breakdown of BM. Can the authors clarify what may be occurring, for example is this evidence of invadopodia formation? This would be very interesting.

Here, the referee is raising another interesting aspect. Indeed, the pioneer cells seem to be active. We can, however, not entirely rule out the involvement of other cells, e.g. POM, in between the margins at this stage of development. Thus, we decided to be cautious with the interpretation of this finding.

Although the bmp embryos are heat shocked, there is no evidence to show whether this is active, e.g. through pSMAD signalling. It would also be useful to see in which optic fissure cells the pSMAD is active in response to bmp.

We previously tested the transgenic line tg(hsp70:bmp4) in a cross to a bmp signaling reporter line, which is activated by canonical bmp signaling via pSmads. A similar approach was previously performed in a similar transgenic line (Heermann et al., 2015) in which the bmp4 was under the control of the rx2 cis regulatory element. For bmp4 we have seen an almost ubiquitous activation within the embryo. Unfortunately, we did not record with a confocal microscope and thus did not record the precise activity within the fissure margins. Due to the time limitation for the revision we were not able to perform this experiment.

The exposure is saturated in the beginning of Movie 5, and the timings are not shown. These should be addressed to enable clearer interpretation of this data.

Priska:

The period of in vivo recording of the data presented in movie 5 is approx. 40 hours. Even though imaging was performed carefully, bleaching occurred. We had to saturate the exposure at the beginning, so that we can observe the end of movie in one file. The timings are implemented into the movie files.

Figure 5. The data suggests that bmp4 prevents separation of the HA/HV, however it is not stated how many embryos this observation was seen in or the total embryos/eyes used for analysis.

We are sorry for not having provided the data previously. The number of embryos used is now included.

The data could also be improved through quantitation for the fluorescent signal against background and could prevent artefacts from mis-aligned optical sections and improve the comparison between bmp and control groups as the qualitative difference appears subtle in the figure presented.

We are not sure what the referee is aiming for with this suggestion. We did not intend to quantify the fluorescent signal. After the identification of the persisting basal lamina in BMP induced embryos, our rational was to analyze the behavior of the POM population after BMP induction, knowing that POM derivatives are important for basal lamina dissolution. To this end we used the sox10 driven lyntdTomato as a readout, induced BMP4 at 26 hpf and started in vivo imaging at 34 hpf. We did not observe any changes in POM cell behavior at the onset of imaging, even though the heat-shock mediated BMP4 induction was preceding 8 hours. However, we observed differences of the POM progeny localization after the “onset of fusion” in the controls. We interpreted these as secondary effects of the hampered fusion.

The term “abnormal hyaloid vein” needs more specific description, e.g. is it “abnormally located”? Can the authors confirm the fli1a:gfp signal observed in Fig6B are hyaloid capillaries and not presumptive HA or HV? There is green signal visible in the centre region of Fig6B - are these proximally located HA/HV or are these distal tissues?

We specified the term in the revised manuscript. The green signal in the center of the optic cup in Figure 6B are hyaloid capillaries. Below them are the hyaloid vein and artery. We indicated this in the revised version.

Line 328. “..site of fusion is eventually displacing the hyaloid vein towards the periphery”. This is a clumsy sentence, while there is also no direct evidence showing this occurring in the study. Can the authors show a sagittal view of this happening to help confirm this?

We performed in vivo imaging to obtain sagittal optical sections of 3D volumes, presented in Figure 6 A-H. The scheme presented in Figure 6 I G is presented in a transverse section to make it more intuitive to understand. We apologize, if this was causing trouble. We now included the orientation in the figure also. We furthermore provide another view to support our conclusion.

Could the heat shock be delayed to arrest the progression of fusion after initiation. This would help show that fusion progression is essential for driving the HV distally, as the HV should be in an intermediate position adjacent to the halted region of fusion?

This is an interesting idea. We did not address whether the initiated fusion process can be stopped by induction of BMP. Such analysis, even though interesting, is beyond the scope of these revisions.

Reviewer 3 Report

The manuscript by Eckert et al. describes an in vivo analysis of optic fissure fusion in zebrafish. The authors describe their classification of pioneer cell mediated fusion of the fissure which is dependent on hyaloid vasculature as a scaffold. The authors also examine the effects of BMP induction on fissure fusion and vasculature organization. They conclude that fissure fusion requires proper interaction of pioneer cells with hyaloid vasculature and that this interaction involves proper regulation/inhibition of BMP signaling within the fissure.

While the authors address important and interesting problems this manuscript adds little new information to this topic while the experiments performed are fraught with technical concerns. Much of the data is simply supporting recent studies showing the association of fissure fusion and hyaloid vasculature. Examination of BMP signaling in this process is interesting but little to no new mechanistic insight is added.

Major Comments:

The authors propose “pioneer cells” as initiators of fissure fusion. While very intriguing, their conclusions are highly overstated and lack several key controls to support this claim. 

1) the pioneer cells are only shown in a 2D image/movie. A 3D reconstruction would better show whether these are in fact pioneers or just represent the first region of the fissure to undergo fusion. It has been previously shown that fusion proceeds from back to front. The authors should also comment on why several recent studies looking into the timing of fissure fusion did not observe the presence of these pioneer cells when examining basal lamina as a readout of fissure fusion. Why do we not see pioneer cells in figure 2H-J?

2) the manuscript lacks detail of the timing of experimental analysis. When do the movies start/end? It is therefore very difficult to grasp what we are looking at and when. Figure 1 states that imaging begins at 37hpf, is that only for G-J? Also, for B-F, if the authors start at completion of morphogenesis, likely ~24hpf, they do not see fusion until 28hpf later, well past what has been reported recently as the timing for zebrafish fissure fusion. This could imply improper environmental control during imaging leading to severe developmental delay?

3) Another overstatement of the data is the plasticity of the pioneer cells. This statement is based simply on the absence of GPF signal from the transgenic line and again shown as a single confocal stack. Furthermore, when examining the supplemental figure associated I am not convinced that the pioneer cells are totally devoid of GFP signal as to make the claim they are rx2 negative. I also point out that a transgenic line is not a direct readout of cellular plasticity. Just because GFP signal is decreased does not indicate that the rx2 protein is no longer active or even reduced. In order to substantiate the plasticity claim the authors need to show a) in situ data for endogenous rx2 as well as IHC for rx2 protein. Furthermore, the timing of rx2 absence from these cells is only a few hours, I am not convinced that in this time rx2 activity would be inhibited enough to confer actually functional differences from other rx2+ cells.

4) evidence for pioneer cell dual potential – NR and RPE, is again lacking. How does one assign flattened morphology from a 2D image?

Association of fissure fusion and hyaloid vessels is not novel and has been shown in several systems. The authors propose these vessels act as scaffolds, again previously hypothesized, but show no direct evidence for this. Perhaps examining what occurs to pioneer cells in the absence of vessels would be the first step? As stated, their conclusions are simply inference from previous literature. 

The authors use a sox10 driven reporter to examine hyaloid vessel development. This transgenic is not an appropriate tool for this experiment. While sox10 expressing cells are known to contribute to hyaloid vasculature, the fli1 or kdrl reporter lines are the best tools for this. In fact, the authors use this line in subsequent experiments but not here. Furthermore, their examination of the sox10 reporter is also very qualitative, no quantification is attempted and no mention of n number of embryos examined is given. The data should also be presented in 3D. 

I am not sure how examining overall vasculature of the eye relates to the process of fissure fusion being examined? BMP induction using hsp is not specific to the fissure and or the eye/vasculature. So I am not certain how to interpret the results that overall vasculature organization is disrupted? What is the significance of this finding in relation to optic fissure? Model diagrams shown are difficult to interpret from the data in the absence of high res 3D reconstructions. Also, in figure 6H, how is it that the control embryos appear to lack vasculature in general?

What is the mechanism of BMP disruption of pioneer cell interaction and disruption of vasculature organization? 

General comments:

The manuscript lacks signs of rigor. There is very little mention of n number of embryos examined or quantification for majority of the data. 

M&M sections that explain critical aspects of the manuscript, in this case time-lapse imaging, should not be referenced to a previous publication. Much of the data is from this technique and in order to best evaluate the outcomes details of how the data was collected is very important. It is not clear if the authors did their time-lapse imaging in 3D and presented as single stack movies, or did only single stack imaging. If only single stacks were collected, this poses a problem as these events should always be examined in 3D due to embryo-embryo variability and difficulty in consistent positioning embryos for imaging.

Imaging data should be presented in 3D. Highly dynamic events such as fissure fusion need to be examined in 3D as single sections can give confounding results, due to axial drift and other imaging issues associated with in vivo imaging, and difficulty in reproducibility. Furthermore, the authors draw several model diagrams but do not show data supporting them, only 2D or max projections.

The manuscript could benefit from more editing. There are several very long sentences and typos. “Fish” should not be used in place of embryos.

Author Response

Dear Reviewer,

at first, we want to thank for the constructive comments on our work, which we highly appreciated. Due to the limited time for revising our work, we mainly focused on the editorial work on our manuscript and the data presentation.

We realized, that the description and presentation of our data needed further improvement. We enclosed the new version of the manuscript, which we carefully revised. The changes are highlighted via the “track changes” modus.

Please find enclosed our answers to your comments, point by point.

The manuscript by Eckert et al. describes an in vivo analysis of optic fissure fusion in zebrafish. The authors describe their classification of pioneer cell mediated fusion of the fissure which is dependent on hyaloid vasculature as a scaffold. The authors also examine the effects of BMP induction on fissure fusion and vasculature organization. They conclude that fissure fusion requires proper interaction of pioneer cells with hyaloid vasculature and that this interaction involves proper regulation/inhibition of BMP signaling within the fissure.

While the authors address important and interesting problems this manuscript adds little new information to this topic while the experiments performed are fraught with technical concerns. Much of the data is simply supporting recent studies showing the association of fissure fusion and hyaloid vasculature. Examination of BMP signaling in this process is interesting but little to no new mechanistic insight is added.

Major Comments:

The authors propose “pioneer cells” as initiators of fissure fusion. While very intriguing, their conclusions are highly overstated and lack several key controls to support this claim. 

the pioneer cells are only shown in a 2D image/movie. A 3D reconstruction would better show whether these are in fact pioneers or just represent the first region of the fissure to undergo fusion. It has been previously shown that fusion proceeds from back to front. The authors should also comment on why several recent studies looking into the timing of fissure fusion did not observe the presence of these pioneer cells when examining basal lamina as a readout of fissure fusion. Why do we not see pioneer cells in figure 2H-J?

The respective movies and images, which are presented in our work, are derived from 3D in vivo recordings over time. Therefore, we were able to visit next to the optical section presented, also the proximal and distal optical sections during analysis. These experiments where performed in vivo. The analysis of the basal lamina degradation on the other hand was performed in fixed embryos. In many recent studies, embryos were also fixed and processed for immunohistochemistry. It is conceivable that this procedure is altering the tissue and thus also the morphology of the margins. This could be a reason why the pioneer cells are maybe not as prominent in such analyses, compared to in vivo recordings. Notably, the label and the temporal dynamics are also important. The dynamics are not represented in fixed samples. Another reason for not seeing pioneer cells, e.g. in Figure 2H-J is that a nuclear labeling was combined with an anti-laminin immunohistochemistry. The membranes were not labeled in these analyses. In the in vivo recordings, however, the membrane labeling was important for the identification of the pioneer cells.

We agree with the referee, that a 3D reconstruction could be helpful. However, many difficulties arise from 3D reconstructions. These are resulting from overlying tissues and differences in fluorescence signal depending on the depth of the tissue during recording. A thorough segmentation and consecutive cropping would be needed in order to obtain a 3D reconstruction which is intuitive for the reader. This would, however, mean intense “image manipulation”. According to our experience the data presentation we provided is currently the best.

the manuscript lacks detail of the timing of experimental analysis. When do the movies start/end? It is therefore very difficult to grasp what we are looking at and when. Figure 1 states that imaging begins at 37hpf, is that only for G-J? Also, for B-F, if the authors start at completion of morphogenesis, likely ~24hpf, they do not see fusion until 28hpf later, well past what has been reported recently as the timing for zebrafish fissure fusion. This could imply improper environmental control during imaging leading to severe developmental delay?

The starting time-points of the movies are given in the movie legends. In the upper left corner of the movie files the recording time is indicated. Naturally, the conditions during imaging are not the same as in an incubator at 28.5 °C in fishmedium. We also revised the movie legends and included the temperature during imaging.

In recent study, it was reported that fusion of the optic fissure starts at approx. 35hpf. (James et al., 2016). This timing is well in line with our observations (Figure1 B-F). Proximal of the optical section we are presenting in Figure 1 the process of fusion is already initiated. When present a more distal section (Figure1 B-F), where fusion starts at around 36 hpf (Figure 1C).

Another overstatement of the data is the plasticity of the pioneer cells. This statement is based simply on the absence of GPF signal from the transgenic line and again shown as a single confocal stack. Furthermore, when examining the supplemental figure associated I am not convinced that the pioneer cells are totally devoid of GFP signal as to make the claim they are rx2 negative. I also point out that a transgenic line is not a direct readout of cellular plasticity. Just because GFP signal is decreased does not indicate that the rx2 protein is no longer active or even reduced. In order to substantiate the plasticity claim the authors need to show a) in situ data for endogenous rx2 as well as IHC for rx2 protein. Furthermore, the timing of rx2 absence from these cells is only a few hours, I am not convinced that in this time rx2 activity would be inhibited enough to confer actually functional differences from other rx2+ cells.

We removed the term plasticity from our manuscript. We observed, that the cells, which establish the cellular contact in between the margins showed a reduced GFP signal up to an almost total loss of GFP signal. We further did not want to imply that rx2 is driving any functional changes, if the referee is suggesting this with the statment: “Furthermore, the timing of rx2 absence from these cells is only a few hours, I am not convinced that in this time rx2 activity would be inhibited enough to confer actually functional differences from other rx2+ cells.”

evidence for pioneer cell dual potential – NR and RPE, is again lacking. How does one assign flattened morphology from a 2D image?

The respective movies and images, which are presented in our work, are derived from 3D in vivo recordings over time. Therefore, we were able to visit next to the optical section presented also the proximal and distal optical sections during analysis. This issue was also discussed above. This mode of analysis enabled us to address the pioneer cell population in 3D over time. The 3D volume was used for the data analysis. We agree, that otherwise data interpretation would be too prone for errors. Also the shape of the cells was addressed using the 3D stack information. Furthermore, a membrane labeling was used to identify the pioneer cells. We apologize that we likely did not stress the technical details sufficiently.

Association of fissure fusion and hyaloid vessels is not novel and has been shown in several systems. The authors propose these vessels act as scaffolds, again previously hypothesized, but show no direct evidence for this. Perhaps examining what occurs to pioneer cells in the absence of vessels would be the first step? As stated, their conclusions are simply inference from previous literature. 

We agree, that also others investigated the role of the blood vessels during optic fissure fusion and that naturally our discussion is including the work and ideas of others.

The pioneer cells, however, were first described by us and we also observed the close interaction with the hyaloid vessel during contact formation. The data was and still is available on bioRxiv and was discussed e.g. by Gestri et al., 2018. It may be that our manuscript was not listed in the reference list, nevertheless it was clearly referred to our prepublished work.

The suggested experiment is indeed a potential first step, however, not feasible in the scope of this revision.

The authors use a sox10 driven reporter to examine hyaloid vessel development. This transgenic is not an appropriate tool for this experiment. While sox10 expressing cells are known to contribute to hyaloid vasculature, the fli1 or kdrl reporter lines are the best tools for this. In fact, the authors use this line in subsequent experiments but not here. Furthermore, their examination of the sox10 reporter is also very qualitative, no quantification is attempted and no mention of n number of embryos examined is given. The data should also be presented in 3D. 

We did not intend to quantify the fluorescent signal. After the identification of the persisting basal lamina in BMP induced embryos, our rational was to analyze the behavior of the POM population after BMP induction, knowing that POM derivatives are important for basal lamina dissolution. To this end we used the sox10 driven lyntdTomato as a readout, induced BMP4 at 26 hpf and started in vivo imaging at 34 hpf. We did not observe any changes in POM cell behavior at the onset of imaging, even though the heat-shock mediated BMP4 induction was preceding 8 hours. However, we observed differences of the POM progeny localization after the “onset of fusion” in the controls. We interpreted these as secondary effects of the hampered fusion.

The referee is right, that we consecutively used a specific marker for blood vessels to further address the changes in vascular architecture.

I am not sure how examining overall vasculature of the eye relates to the process of fissure fusion being examined? BMP induction using hsp is not specific to the fissure and or the eye/vasculature. So I am not certain how to interpret the results that overall vasculature organization is disrupted? What is the significance of this finding in relation to optic fissure? Model diagrams shown are difficult to interpret from the data in the absence of high res 3D reconstructions. Also, in figure 6H, how is it that the control embryos appear to lack vasculature in general?

The referee is correct in that there is no obvious connection between overall vasculature of the optic cup and optic fissure fusion. However, we became interested in the vasculature after observing that the hyaloid vein is at first located in the proximal optic fissure and moves distally during fusion, creating the VRV in the process. Kaufman et al. (2015) describe that the ventral part of the superficial annular vessel is derived from the VRV. Since the VRV is not properly forming in our BMP induced embryos, we believe it is likely that the HV failing to move distally causes the ventral part of the annular vessel to be missing. In our manuscript, we also mentioned smaller malformations of ocular vasculature (such as the bifurcation of the DRV) for sake of completeness, but we do not strive to explain them. We agree with the referee that it would be very difficult to do so. We rewrote parts of the paragraph referring to Figure 6 in order to better explain our motivation and findings.

The control embryo depicted in figure 6E-H does not lack vasculature. The optical plane shown in 6H only includes few vessels, as it is a very distal plane.

What is the mechanism of BMP disruption of pioneer cell interaction and disruption of vasculature organization? 

General comments:

The manuscript lacks signs of rigor. There is very little mention of n number of embryos examined or quantification for majority of the data. 

We apologize for this. We carefully revised the manuscript and included the number of embryos included.

M&M sections that explain critical aspects of the manuscript, in this case time-lapse imaging, should not be referenced to a previous publication. Much of the data is from this technique and in order to best evaluate the outcomes details of how the data was collected is very important. It is not clear if the authors did their time-lapse imaging in 3D and presented as single stack movies, or did only single stack imaging. If only single stacks were collected, this poses a problem as these events should always be examined in 3D due to embryo-embryo variability and difficulty in consistent positioning embryos for imaging.

We agree with the referee, that recordings in 2D would be problematic. We thus performed 3D in vivo imaging over time. We extended the Materials and Methods section and now provide the full description of our imaging experiments.

Imaging data should be presented in 3D. Highly dynamic events such as fissure fusion need to be examined in 3D as single sections can give confounding results, due to axial drift and other imaging issues associated with in vivo imaging, and difficulty in reproducibility. Furthermore, the authors draw several model diagrams but do not show data supporting them, only 2D or max projections.

We agree with the referee, that a 3D reconstruction could be helpful. However, many difficulties arise from 3D reconstructions. These are resulting from overlying tissues and differences in fluorescence signal depending on the depth of the tissue during recording and also from reconstruction algorithm.

We agree that data analysis based on 2D imaging would be problematic. We thus examine our time-lapse data in 3D stacks.

The manuscript could benefit from more editing. There are several very long sentences and typos. “Fish” should not be used in place of embryos.

The manuscript was carefully revised.

Round 2

Reviewer 1 Report

This manuscript has been editorially improved in response to the reviewers' comments.  However, experimentally, it remains virtually unchanged.  Many of the reviewers' comments are simply set aside by saying that a response is beyond the scope of the study, or that the point is interesting and merits further investigation.   

Reviewer 3 Report

I appreciate the authors attempts to edit their manuscript within the short time period provided. However, the revised version is still frought with overstatements, poor writing and key missing experiments. I feel my concerns were not addressed sufficiently. In light of the limited attempt to revise the manuscript I stand by my original decision.

Personally I found the time line expectations of the journal to be significantly too short to warrant any meaningful revision to the work. One month is not enough time to address the major shortcomings of this work and is frankly unfair to the authors. I would suggest the authors work on the suggested revisions from both reviewers and re-submit the work at a later time. Their findings are still interesting and impactful, however, the current state of their mansucript does not warrant publication.